# RDT-1B: a Diffusion Foundation Model for Bimanual Manipulation

**Songming Liu**[*], **Lingxuan Wu**[*], **Bangguo Li**, **Hengkai Tan**, **Huayu Chen**,
**Zhengyi Wang**, **Ke Xu**, **Hang Su**[†], **Jun Zhu**[†]
[1]Department of Computer Science & Technology, Institute for AI, BNRist Center,
Tsinghua-Bosch Joint ML Center, THBI Lab, Tsinghua University

## Abstract

Bimanual manipulation is essential in robotics, yet developing foundation models is extremely challenging due to the inherent complexity of coordinating two robot arms (leading to multi-modal action distributions) and the scarcity of training data. In this paper, we present the Robotics Diffusion Transformer (RDT), a pioneering diffusion foundation model for bimanual manipulation. RDT builds on diffusion models to effectively represent multi-modality, with innovative designs of a scalable Transformer to deal with the heterogeneity of multi-modal inputs and to capture the nonlinearity and high frequency of robotic data. To address data scarcity, we further introduce a Physically Interpretable Unified Action Space, which can unify the action representations of various robots while preserving the physical meanings of original actions, facilitating learning transferrable physical knowledge. With these designs, we managed to pre-train RDT on the largest collection of multi-robot datasets to date and scaled it up to $1.2$B parameters, which is the largest diffusion-based foundation model for robotic manipulation. We finally fine-tuned RDT on a self-created multi-task bimanual dataset with over 6K+ episodes to refine its manipulation capabilities. Experiments on real robots demonstrate that RDT significantly outperforms existing methods. It exhibits zero-shot generalization to unseen objects and scenes, understands and follows language instructions, learns new skills with just 1∼5 demonstrations, and effectively handles complex, dexterous tasks. We refer to the project page for the code and videos.

## 1 Introduction

Bimanual manipulation is essential for robots to accomplish real-world tasks (Edsinger & Kemp, 2007). For practical applications, a useful manipulation policy should be able to generalize to unseen scenarios, such as unseen objects and scenes. However, current approaches either depend on task-specific primitives (Mirrazavi Salehian et al., 2017; Rakita et al., 2019; Grannen et al., 2023a) or are limited to small-scale model, data and simple tasks (Krebs et al., 2021; Franzese et al., 2023; Grannen et al., 2023b; Zhao et al., 2023; Grotz et al., 2024; Liu et al., 2024), thereby exhibiting only narrow generalization and failing in complex tasks. Following the success in natural language processing (Achiam et al., 2023; Touvron et al., 2023) and computer vision (Radford et al., 2021; Kirillov et al., 2023), one promising direction to enable generalizable behaviors is to develop a foundation model through imitation learning on large-scale datasets.

Developing bimanual manipulation foundation models confronts the dual challenges of data scarcity and architectural limitations. The prohibitive costs of dual-arm systems create severe data scarcity (Sharma et al., 2018; Collaboration et al., 2023), fundamentally conflicting with the data-hungry nature of foundation models. Inspired by recent attempts in unimanual manipulation (Brohan et al., 2023; Kim et al., 2024), we mitigate this through cross-robot pretraining: leveraging multi-robot datasets for pre-training followed by target-robot fine-tuning, amplifying data volume by three orders of magnitude to extract transferable physical priors. However, two interconnected technical barriers emerge. First, the doubled action space induces multi-modal action distributions (Li, 2006; Jia et al., 2024) (see Fig. 2b for an illustrative example) that demand *expressiveness* capability beyond current

---

[*]Equal contribution; [†]Corresponding authors at {suhangss, dcszj}@tsinghua.edu.cn

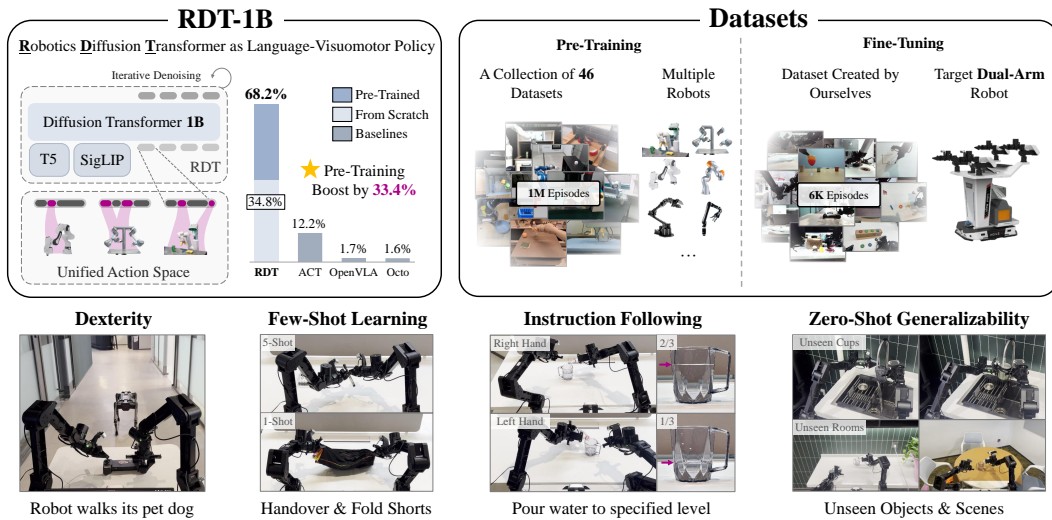

Figure 1: **Overview of Robotics Diffusion Transformer with 1B-Parameters (RDT-1B)**, a language-conditioned visuomotor policy for bimanual manipulation, with state-of-the-art generalizability to unseen scenarios (See App. H for metric calculation details).

methods (Zhao et al., 2023; Brohan et al., 2023; Kim et al., 2024), while simultaneously requiring *scalabilty* for stable large-scale training on multimodal data (text, vision, actions). Beyond architectural constraints, physical and action space variations across robots introduce data heterogeneity that risks negative transfer (Pan & Yang, 2009). Existing solutions either discard robots with structural inconsistencies or retain only cross-robot invariant features (Brohan et al., 2023; Ghosh et al., 2023; Shah et al., 2023a), sacrificing valuable data diversity essential for generalization.

In this paper, we introduce the *Robotics Diffusion Transformer (RDT)*, the largest bimanual manipulation foundation model with strong generalizability. RDT employs diffusion transformer (DiT) as its scalable backbone (Peebles & Xie, 2023), with special designs for language-conditioned bimanual manipulation. For expressiveness, RDT excels in capturing the full modalities of bimanual actions from massive data by using the capacity of diffusion models to represent complex distributions (Sohn et al., 2015; Ho et al., 2020). For scalability, we harness the Transformer backbone and carefully design the multi-modal encoding to eliminate the heterogeneity of various modalities. Moreover, robotic data is differed significantly from images and videos with temporal and spatial continuity (Chen et al., 2019; Liang et al., 2022). To characterize its inherent nonlinear dynamics (de Wit et al., 2012), high-frequency changes (Ghosh et al., 2023), and the unstable numerical range, we make important modifications to the original DiT structure, including MLP decoding, improved normalization, and alternate injection of conditions (see Fig. 4 for their importance). To further enable training RDT on heterogeneous data, we propose the *Physically Interpretable Unified Action Space*, a unified action format for various robots with gripper arms. This innovative format mitigates potential conflicts between different robots while retaining the physical meanings of the original actions, which can promote the model to learn generalizable physical knowledge across diverse robotic datasets.

With the above designs, we managed to pre-train the RDT model on the largest collection of multi-robot datasets to date (Collaboration et al., 2023; Walke et al., 2023; Fang et al., 2023; Kumar et al., 2024) and scale it up to 1.2B parameters, which is the largest diffusion-based pre-trained model for robotic manipulation. To further enhance its bimanual manipulation capabilities, we fine-tuned the RDT on a self-collected multi-task bimanual dataset comprising over 6K+ trajectories, which is one of the most extensive bimanual datasets. In our experiments, we have comprehensively evaluated RDT against strong baselines in both bimanual manipulation and robotic foundation models. Results show that RDT achieves state-of-the-art performance, outperforming baselines by achieving an improvement of 56% in success rates across a wide spectrum of challenging tasks. In particular, RDT has exceptional zero-shot and few-shot ($1 \sim 5$ shots) generalizability to unseen objects, scenes, instructions, and even skills. RDT is also capable of accomplishing tasks requiring fine-grained operations, such as controlling a robot dog with a joystick. Finally, ablation studies show that diffusion modeling, large model size, and large data size all contribute to superior performance.

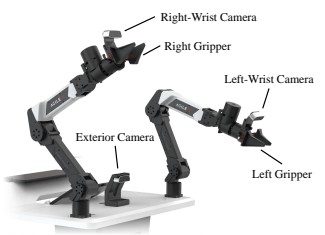 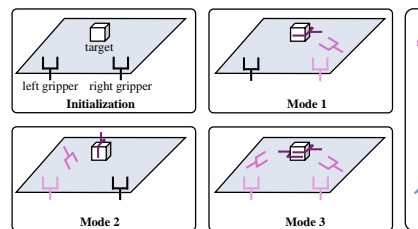 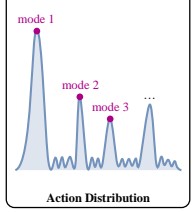

(a) Target dual-arm robot        (b) Illustration of multi-modality

Figure 2: **(a)** Schematic diagram of the ALOHA dual-arm robot. **(b)** A toy example of grasping a cube. Compared with unimanual manipulation, bimanual manipulation has more possible action modes, leading to stronger multi-modality. Colors from light to dark indicate that time goes forward.

## 2 RELATED WORK

**Learning-based Bimanual Manipulation.** One substantial challenge in learning a bimanual manipulation policy is the high dimensionality of the action space, which exacerbates the data scarcity (Zollner et al., 2004; Smith et al., 2012; Lioutikov et al., 2016; Stepputtis et al., 2022) and the multi-modal behavior (Colomé & Torras, 2018; 2020; Figueroa & Billard, 2017; Sharma et al., 2018; Xie et al., 2020; Franzese et al., 2023). Some works have developed more cost-effective interfaces for data collection (Zhao et al., 2023; Aldaco et al., 2024), but they are limited to specific hardware configurations and still insufficient to bridge the data gap for a generalizable policy. Others attempt to reduce data requirements by introducing inductive biases, such as distinguishing two arms for stabilization and functionality (Grannen et al., 2023b), parameterizing movement primitives (Batinica et al., 2017; Amadio et al., 2019; Chitnis et al., 2020; Franzese et al., 2023), or using voxel representations (Grotz et al., 2024; Liu et al., 2024). These methods use strong priors or simplified modeling, which successfully reduce the action space, but at the cost of a reduced scope of application and inability to express the multi-modality of bimanual behaviors (Pearce et al., 2023).

**Foundation Models for Robotics.** Foundation models have shown immense promise in enabling generalizable behaviors by training multi-task "generalist" models (Brohan et al., 2022; 2023; Ghosh et al., 2023; Kim et al., 2024) on large multi-task robot datasets (Collaboration et al., 2023; Brohan et al., 2022; Fang et al., 2023). Most studies adapt large vision-language models to directly predict action (Brohan et al., 2022; Driess et al., 2023; Brohan et al., 2023; Collaboration et al., 2023; Kim et al., 2024). While demonstrating generalization to new objects and tasks, they face issues with quantization errors and uncoordinated behaviors (Pearce et al., 2023) when applied to bimanual manipulation. It's largely due to their discretization of action spaces. To enhance precision, diffusion models have been used for continuous control (Ho et al., 2020; Chi et al., 2023; Pearce et al., 2023; Ghosh et al., 2023). Ghosh et al. (2023) pre-train a Transformer-based diffusion policy on a subset of Open X-Embodiment (Collaboration et al., 2023) dataset (25 datasets), with up to 93M parameters.

## 3 PROBLEM FORMULATION AND CHALLENGES

We start by formulating the task and elaborating on the challenges. To evaluate the model on the hardware, we choose the ALOHA dual-arm robot as our target robot since it is one of the most representative dual-arm robots and is suitable for collecting human demonstration data via teleoperation (Zhao et al., 2023; Fu et al., 2024; Aldaco et al., 2024). Fig. 2a shows a schematic diagram of the target robot, which consists of two arms with grippers and three cameras. Note that our setting and foundation model are generic to any dual-arm gripper robot.

We consider the concrete task of language-conditioned bimanual manipulation with vision, which is fundamental in robotics and has great value in real-world scenarios such as household (Stepputtis et al., 2020; Brohan et al., 2022; Zhao et al., 2023). Formally, given a language instruction $\ell$, the policy is presented with an observation $o_t$ at time $t \in \mathbb{N}^+$; and then it produces an action $a_t$ to control *two robot arms* to achieve the goal specified by $\ell$. The observation is represented as a triple $o_t := (X_{t-T_{\text{img}}+1:t+1}, z_t, c)$, where $X_{t-T_{\text{img}}+1:t+1} := (X_{t-T_{\text{img}}+1}, \dots, X_t)$ is the RGB observation history of size $T_{\text{img}}$, $z_t$ is the low-dimensional proprioception of the robot, and $c$ is the control frequency. The action $a_t$ is usually a subset of the desired proprioception $z_{t+1}$[1].

---

[1]E.g., $z_t$ may include the gripper position at time $t$, and $a_t$ can be the target gripper position at step $t + 1$.

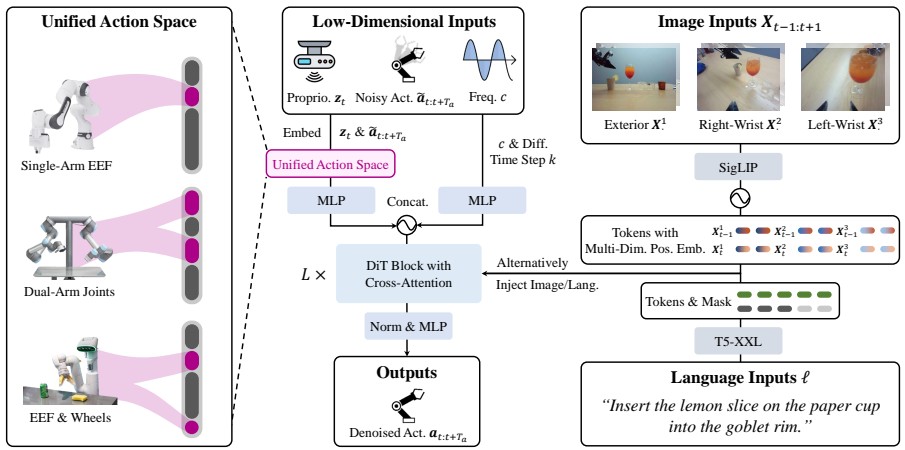

Figure 3: **RDT framework.** Heterogeneous action spaces of various robots are embedded into a unified action space for multi-robot training. **Inputs:** proprioception $z_t$, noisy action chunk $\tilde{a}_{t:t+T_a}$, control frequency $c$, and diffusion time step $k$, acting as denoising inputs; image inputs ($T_{\text{img}} = 2$ and $X. = \{X^1, X^2, X^3\}$ denotes a set of images from exterior, right-wrist, and left wrist cameras) and language inputs, acting as conditions. **Outputs:** denoised action chunk $a_{t:t+T_a}$.

A specific task in bimanual manipulation typically consists of multiple elements: a *skill* (e.g., verbs like "pick", "wipe", or "open"), an *object* (e.g., nouns like "bottle", "table", or "door"), a *scene* (i.e. the environment in which the task takes place), and a *modality* describing how the skill is performed (e.g., adverbials like "pick the bottle with the left hand"). When encountering a new task, a practical policy is required to generalize to unseen[2] elements in the task, which is particularly challenging for previous rule-based methods (Mirrazavi Salehian et al., 2017; Rakita et al., 2019; Grannen et al., 2023a) as well as learning-based methods that are limited to either small models and data or simple tasks, as discussed in Sec. 2.

We aim to train a foundation model policy via imitation learning to achieve generalizability. However, the available data for a specific dual-arm robot is particularly scarce ($< 10\text{K}$ trajectories) due to high hardware costs, far from the common requirement to train a foundation model. To address this, we propose to employ a pre-training and fine-tuning pipeline (Radford et al., 2018) to take advantage of data from multiple robots by drawing inspiration from recent advances in unimanual manipulation (Ghosh et al., 2023; Collaboration et al., 2023; Kim et al., 2024). In this manner, we would expand the data size by three orders of magnitude. Specifically, we first pre-train the model on a large-scale multi-robot dataset $\mathcal{D}_{\text{pre}}$ (mostly single-arm) and then fine-tune on a dataset of the target robot $\mathcal{D}_{\text{ft}}$. We denote the dataset by $\mathcal{D}. = \{(\ell^{(i)}, o_t^{(i)}, a_t^{(i)}) \mid 0 \leq t < T^{(i)}, 1 \leq i \leq N\}$, where $T^{(i)}$ is the length of the $i$-th trajectory and $N$ is the number of trajectories. Moreover, it is worth emphasizing that our goal is to use multi-robot data to enhance the model's generalizability in bimanual manipulation *rather than* developing a cross-embodiment model for various robots. There are two main challenges to developing such a foundation model with multi-robot data:

**Challenge 1: How to design a powerful architecture?** A generalizable foundation model necessitates a powerful architecture. This requirement encompasses two primary aspects. Firstly, the architecture must possess sufficient *expressiveness* to capture the multi-modality in the action distribution. Fig. 2b illustrates a toy example where the robot attempts to grasp a cube. We can see that there are many modes to finish this task, in contrast to unimanual manipulation, where only one robot arm is controlled. When collecting demonstrations, the human operator may randomly pick one of them, leading to multi-modality in the collected action data. Secondly, *scalability* is necessary for such an architecture. As a foundation model, it should effectively process heterogeneous inputs from various modalities (text, images, actions, etc.) while being scalable to train stably on large datasets.

**Challenge 2: How to train on heterogeneous data?** Training on multi-robot data presents a unique challenge of data heterogeneity. The physical structure and the action space can vary greatly across different robots. Previous attempts either restrict themselves to a subset of robots with similar action spaces (Yang et al., 2023; Ghosh et al., 2023; Kim et al., 2024) or only retain a subset of inputs

---

[2]*unseen* means that a certain element has not appeared in the training data.

sharing the same structure (Collaboration et al., 2023; Yang et al., 2024), at the cost of losing a lot of information. It remains largely under-addressed on how to train models on such heterogeneous data.

# 4 ROBOTICS DIFFUSION TRANSFORMER

We now present Robotics Diffusion Transformer (RDT), as illustrated in Fig. 3. In Sec. 4.1, we present the diffusion model and the corresponding architecture to address Challenge 1. In Sec. 4.2, we resolve Challenge 2 by proposing a physically interpretable unified action space to unify various robot action spaces and enable multi-robot pre-training. We also collect a comprehensive multi-task bimanual dataset for fine-tuning to improve the bimanual manipulation capabilities of RDT.

## 4.1 RDT MODEL

**Diffusion Modeling.**  Due to multi-modality, given the language instruction $\ell$ and observation $\boldsymbol{o}_t$, there may be many possible actions $\boldsymbol{a}_t$ to proceed with the task. The policy will learn the "average" of action modes if we model it as a deterministic mapping $(\ell, \boldsymbol{o}_t) \mapsto \boldsymbol{a}_t$ and regress the tuples of $(\ell, \boldsymbol{o}_t, \boldsymbol{a}_t)$ in the training data. This may result in out-of-distribution actions, such as the arithmetic mean of multiple modes, which can be completely infeasible (Pearce et al., 2023). Instead, we choose to model the continuous conditional distribution $p(\boldsymbol{a}_t|\ell, \boldsymbol{o}_t)$. As discussed in Sec. 2, among various approaches, diffusion models excel in both expressiveness and sampling quality, but can be slow to sample high-dimensional data (e.g., images). Luckily, for our settings, the drawback is minor since that $\boldsymbol{a}_t$ has a much lower dimension than images, which requires only minimal sampling overhead. This has made diffusion models an ideal choice for policy as in Chi et al. (2023).

Nevertheless, employing diffusion models for robotic tasks faces unique challenges since the inherent properties of robotic physics quantities (i.e., the action and proprioception) are different from image/video data. Image and video data, while high-dimensional, often exhibit a degree of temporal and spatial continuity (Chen et al., 2019; Liang et al., 2022), with changes between frames typically being incremental. In contrast, robotic physics quantities are characterized by its *nonlinear dynamics* (de Wit et al., 2012) and the potential for *high-frequency changes* stemming from the physical interactions, such as collision, constraints, and material properties like damping. Moreover, the quantities also feature an *unstable numerical range*, probably due to extreme values caused by unreliable sensors. This underscores the necessity of adapting current diffusion models to effectively capture the instability and nonlinearity of robot data. Next, we will first elaborate on diffusion formulation and then introduce our design of architecture to resolve these challenges.

When making a decision with diffusion policies, we first sample a totally noisy action $\boldsymbol{a}_t^K \sim \mathcal{N}(\boldsymbol{0}, \boldsymbol{I})$ and then perform $K \in \mathbb{N}^+$ denoising steps to denoise it to a clean action sample $\boldsymbol{a}_t^0$ from $p(\boldsymbol{a}_t|\ell, \boldsymbol{o}_t)$:

$$\boldsymbol{a}_t^{k-1} = \frac{\sqrt{\bar{\alpha}^{k-1}}\beta^k}{1 - \bar{\alpha}^k}\boldsymbol{a}_t^0 + \frac{\sqrt{\alpha^k}(1 - \bar{\alpha}^{k-1})}{1 - \bar{\alpha}^k}\boldsymbol{a}_t^k + \sigma^k \boldsymbol{z}, \quad k = K, \ldots, 1, \tag{1}$$

where $\{\alpha^k\}_{k=1}^K, \{\sigma^k\}_{k=1}^K$ are scalar coefficients pre-defined by a noise schedule (Nichol & Dhariwal, 2021). Here, $\beta^k := 1 - \alpha^k$, and $\bar{\alpha}^{k-1} := \prod_{i=1}^{k-1} \alpha^i, \boldsymbol{z} \sim \mathcal{N}(\boldsymbol{0}, \boldsymbol{I})$ if $k > 1$, else $\bar{\alpha}^{k-1} = 1, \boldsymbol{z} = \boldsymbol{0}$. However, $\boldsymbol{a}_t^0$ is intractable before sampling is finished. We opt to use a learnable denoising network $f_{\boldsymbol{\theta}}$ with parameters $\boldsymbol{\theta}$ to estimate the clean sample from a noisy one: $\boldsymbol{a}_t^0 \leftarrow f_{\boldsymbol{\theta}}(\ell, \boldsymbol{o}_t, \boldsymbol{a}_t^k, k)$. To train such a network, we will minimize the following mean-squared error (MSE) of denoising:

$$\mathcal{L}(\boldsymbol{\theta}) := \text{MSE}\left(\boldsymbol{a}_t, f_{\boldsymbol{\theta}}(\ell, \boldsymbol{o}_t, \sqrt{\bar{\alpha}^k}\boldsymbol{a}_t + \sqrt{1 - \bar{\alpha}^k}\boldsymbol{\epsilon}, k)\right), \tag{2}$$

where $k \sim \text{Uniform}(\{1, \ldots, K\})$, $\boldsymbol{\epsilon} \sim \mathcal{N}(\boldsymbol{0}, \boldsymbol{I})$, and $(\ell, \boldsymbol{o}_t, \boldsymbol{a}_t)$ is sampled from our training dataset. Later in this paper, we will denote noisy action inputs by $\tilde{\boldsymbol{a}}_t := \sqrt{\bar{\alpha}^k}\boldsymbol{a}_t + \sqrt{1 - \bar{\alpha}^k}\boldsymbol{\epsilon}$, in which the superscript of $k$ is dropped for simplicity. Besides, in practice, we prefer to predict a sequence of actions, i.e., an action chunk, in one shot to encourage temporal consistency (Chi et al., 2023) and to alleviate error accumulation over time by reducing number of decisions in a task (Zhao et al., 2023). Specifically, we model $p(\boldsymbol{a}_{t:t+T_a}|\ell, \boldsymbol{o}_t)$, where $\boldsymbol{a}_{t:t+T_a} := (\boldsymbol{a}_t, \ldots, \boldsymbol{a}_{t+T_a-1})$ is an action chunk and $T_a$ denotes the chunk size (Zhao et al., 2023). We provide a detailed discussion in App. A.

We now present the design of the architecture, including the encoding of multi-modal inputs and the network structure of $f_{\boldsymbol{\theta}}$, while details are deferred to App. B.

**Encoding of Heterogeneous Multi-Modal Inputs.** The heterogeneity of multi-modal inputs is reflected in the structure; that is, the format and number of dimensions of each modality are significantly different. This has posed challenges for mult-modal training. To address this, we encode these diverse modalities into a unified latent space. Below are the encoding methods:

- **Low-Dimensional Inputs** are low-dimensional vectors that represent physical quantities of the robot, including the proprioception, the action chunk, and the control frequency. To encode them, we use MLPs (with Fourier features (Tancik et al., 2020)), which can effectively capture the *high-frequency changes* in low-dimensional spaces.

- **Image Inputs** are high-dimensional and contain rich spatial and semantic information. To extract compact representations, we use an image-text-aligned pre-trained vision encoder, SigLIP (Zhai et al., 2023). We fix its weights during training to save GPU memory.

- **Language Inputs** are of varying length and highly abstract, posing integration challenges due to their complexity and ambiguity. To encode them, we use a pre-trained Transformer-based language model, T5-XXL (Raffel et al., 2020). We also fix its weights during training to save GPU memory.

Besides, heterogeneity also manifests in both cross-modal and intra-modal information density. First, modalities differ inherently in information capacity (e.g., visual inputs typically yield more tokens than text). Second, substantial variance exists within modalities, as seen in robotic perception where exterior cameras capture broader scenes versus the limited views from wrist cameras (Fig. 3). This discrepancy risks shortcut learning: focusing on exterior views while neglecting details from wrist cameras. To promote balanced multimodal integration, we implement stochastic independent masking across modalities during encoding, preventing overreliance on specific inputs.

**Network Structure of $f_\theta$.** We choose Transformer as the scalable backbone network (Bao et al., 2023; Peebles & Xie, 2023) and make the following three key modifications from Diffusion Transfomer (DiT) by considering the characteristics of our robotic problem:

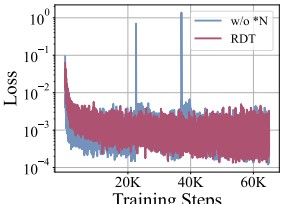

(a) Loss w/o QKN & RMSN

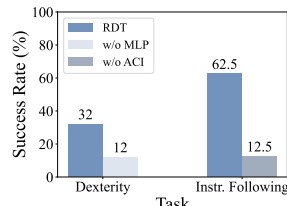

(b) Task w/o MLP or ACI

Figure 4: **(a)** Unstable loss curve during training without QKNorm & RMSNorm. **(b)** Success rates of RDT (w/o MLP Decoder or w/o ACI) in tasks of *Robot Dog* (walk straight sub-task) and *Pour Water-L-1/3* (correct amount sub-task). See Fig. 5 for task definitions. All the models are without pre-training in this experiment due to resource constraints.

- **QKNorm & RMSNorm.** The *unstable numerical range* of the inputting robotic physical quantities can lead to problems such as gradient instability and numerical overflow, especially when training large foundation models. To solve this problem, we add QKNorm (Henry et al., 2020) to avoid numerical instability when calculating attention. Besides, we also note that our problem can be considered as a time series forecasting task, and the centering operation in the original DiTs' LayerNorm could cause *token shift* and *attention shift*, thus destroying the symmetry of the time series (Huang et al., 2024). Therefore, we replace LayerNorm with RMSNorm (Zhang & Sennrich, 2019) without a centering operation. Fig. 4a shows that large-scale pre-training tends to be very unstable or even explode without this modification.

- **MLP Decoder.** To improve the approximation capability for *nonlinear* robot actions, we replace the final linear decoder with a nonlinear MLP decoder as a projection from the latent space back to the physical space. As empirically shown in Fig. 4b, without this design, RDT cannot effectively capture nonlinear dynamics and thus loses the ability to accomplish dexterous tasks that require delicate operations.

- **Alternating Condition Injection (ACI).** In our model, image and language inputs serve as conditions, which are high-dimensional and variable in length, contrasting with the class label conditions in traditional DiTs (Peebles & Xie, 2023). These informative conditions are challenging to compress into a single token, making the original adaptive layer norm approach unsuitable. Therefore, we employ cross-attention to accommodate conditions of varying lengths avoiding the information loss in further compression. Besides, we further analyze that, given that image tokens are usually much more than text tokens, simultaneous injection of both modalities tends to overshadow text-related information, thus impairing the capability of the instruction following (see Fig. 4b for quantitative results). To

mitigate this issue, we strategically alternate between injecting image and text tokens in successive layers' cross-attention rather than injecting both in every layer.

## 4.2 DATA

**Training on Heterogeneous Multi-Robot Data.**   To enable training on heterogeneous multi-robot data, we need a unified action space shared among various robots to provide a unified format for multi-robot actions. The mapping from the original action space of a robot to the unified action space should be physically interpretable, and each dimension of the space should have a clear physical meaning. This can encourage the model to learn shared physical laws from different robot data, thereby improving the efficiency of learning from data of different robots (Shah et al., 2023a).

The design of the space consists of two steps. Firstly, for each robot, we can use a single space to accommodate both its proprioception $z_t$ and action $a_t$. This is because $a_t$ is usually a subset of the desired $z_{t+1}$ (de Wit et al., 2012; Kouvaritakis & Cannon, 2016), and thus the space of $z_t$ naturally contains the space of $a_t$. Secondly, we design a unified space that encompasses all the main physical quantities of most robots with gripper arms. As illustrated in the left side of Fig. 3, we embed the action space of a robot into this unified space by filling each element of the original action vector into the corresponding position of the unified action space vector according to its physical meaning, with the remaining positions being padded. The specific definition of the space is given in App. C.

With this unified space, we are able to pre-train RDT on data from almost all modern robots with gripper arms, and greatly expand the data scale towards the requirement for a foundation model. Specifically, our collection of pre-training datasets includes 46 datasets of various robots, with a total size of 1M+ trajectories and 21TB. More details and preprocessing are deferred to App. D.

**Collecting a Comprehensive Multi-Task Bimanual Dataset.**   Though having been pre-trained on large-scale datasets, RDT could still need help to zero-shot generalize to the target dual-arm robot due to the embodiment gap. To bridge the gap, we need to collect a multi-task bimanual dataset on the target robot for fine-tuning. Recent advances in large language models (Ziegler et al., 2019; Brown et al., 2020; Touvron et al., 2023) have shown that high-quality fine-tuning datasets are crucial for model performance. We ensure the high quality of our dataset from three aspects: **(1)** Regarding quantity, we have collected 6K+ trajectories, making our dataset one of the largest bimanual datasets nowadays; **(2)** Regarding comprehensiveness, we consider 300+ challenging tasks, covering most manipulation task types, from pick-and-place to plugging cables, even including writing math equations; **(3)** Regarding diversity, we prepare 100+ objects with rigid and non-rigid bodies of various sizes and textures and 15+ different rooms with different lighting conditions. Besides, we further utilize GPT-4-Turbo (Achiam et al., 2023) to rewrite human-annotated instructions to increase text diversity. For more information, we refer to Fig. 6 and App. E.

## 5 EXPERIMENTS

We aim to answer the following questions through real-robot experiments: $\mathcal{Q}$1: Can RDT zero-shot generalize to unseen objects and scenes? $\mathcal{Q}$2: How effective is RDT's zero-shot instruction-following capability for unseen modalities? $\mathcal{Q}$3: Can RDT facilitate few-shot learning for previously unseen skills? $\mathcal{Q}$4: Is RDT capable of completing tasks that require delicate operations? and $\mathcal{Q}$5: Are large model sizes, extensive data, and diffusion modeling helpful for RDT's performance?

## 5.1 EXPERIMENT SETUPS

**Tasks.**   We select 7 challenging tasks to evaluate the generalizability and capabilities of RDT from different dimensions, including complex scenarios that the model may encounter in real-world tasks, such as various unseen elements and dexterous manipulation. An illustration of the dimension of each task is given in Table 1 while detailed definitions and visualizations are provided in Fig. 5.

**Data.**   We use the pre-training and fine-tuning datasets in Sec. 4.2. We now list the number of demos related to each task in our fine-tuning dataset. *Wash Cup*: 133 demos for seen cups combined and 0 demos for unseen cups; *Pour Water*: 350 demos for seen rooms combined and 0 demos for unseen

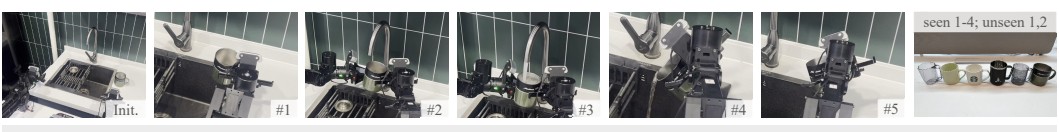

***Wash Cup:*** *"Fill the mug with water from the faucet and pour it into the sink."* The mug is randomized up to 10cm. The robot needs to Pick Up Cup (#1), Turn On Faucet (#2), Get Water (#3, to ensure that the water falls into the cup), Pour Out Water (#4), and Place Back Up (#5). The last image shows seen and unseen cups (from left to right).

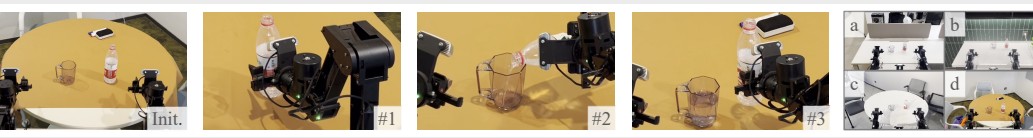

***Pour Water:*** *"Pour water from the bottle into the mug."* The bottle and mug are randomized up to 6cm. The robot needs to Pick Up Bottle (#1), Pour Water (#2), and Place Back Bottle (#3). The last image shows one seen room (a) and three unseen rooms (b-d).

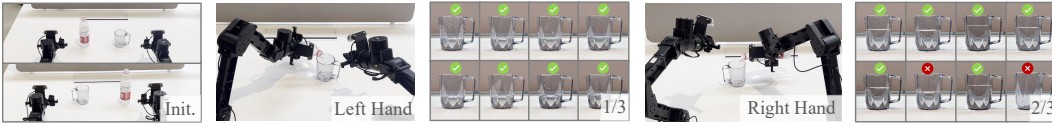

***Pour Water-L-1/3 (-R-2/3):*** *"Pour water from the bottle into the mug until about one-third (two-thirds) with the left (right) hand."* In the left-hand task, the bottle is initially placed to the left of the mug, and vice versa. The bottle and mug are randomized up to 6cm.

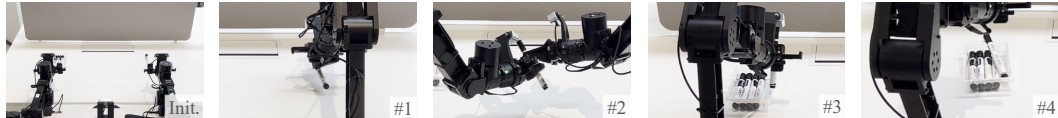

***Handover:*** *"Pick up the black marker on the right and put it into the packaging box on the left."* The marker pen and box are randomized up to 3cm. The robot needs to Pick Up Pen (#1), Switch Hand (#2), Drop Pen (#3), and ensure it can Fall into Box (#4).

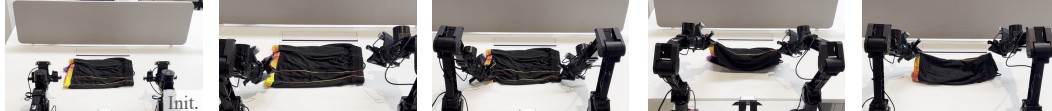

***Fold Shorts:*** *"Fold the basketball shorts into a rectangle."* The shorts are randomized up to 3cm and wrinkles are also randomized.

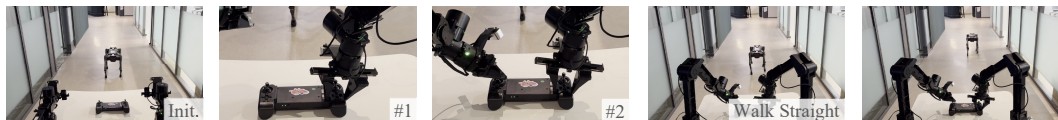

***Robot Dog:*** *"Push the left joystick forward to make the robot dog walk straight forward."* The remote control is randomized up to 4cm and the robot dog is up to 50cm. The robot needs to Grab Remote (#1) and Push Joystick (#2) as straight as possible to make the robot dog walk in a straight line as shown in the last two images.

Figure 5: **Task definitions and visualizations.** For 7 challenging tasks, we describe their language instruction, randomization, and definitions of each sub-task. For *Pour Water-L-1/3* and *Pour Water-R-2/3*, we show the resulting water levels in two images.

rooms; *Pour Water-L-1/3* & *Pour Water-R-2/3*: 18 demos for the water level of little, 19 demos for half, and 19 demos for full; *Handover*: 5 demos; *Fold Shorts*: 1 demo; *Robot Dog*: 68 demos.

**Model Training and Inference.** We scale the size of RDT up to 1.2B parameters, establishing it as the currently largest diffusion-based robotic foundation model. The model is pre-trained on 48 H100 80GB GPUs for a month, giving a total of 1M training iteration steps. It takes three days to fine-tune this model using the same GPUs for 130K steps. We defer further details to App. F, including the running platform, design choices, and data augmentation techniques. For real-time inference, we adopt DPM-Solver++ (Lu et al., 2022), a recent sampling accelerator of diffusion models. It can reduce the diffusion steps required to sample an action chunk from 100 steps to 5 steps, achieving an action chunk inference frequency of 6 Hz (action chunks per second) and an average action inference frequency of 381 Hz (actions per second) on the target robot's onboard RTX 4090 24GB GPU.

**Baselines.** To comprehensively evaluate RDT, we consider the most advanced baselines in robotic foundation models and bimanual manipulation, including Action Chunking with Transformers

Table 1: **Dimensions when designing tasks.** For *Pour Water-L-1/3* and *Pour Water-R-2/3*, only the water levels of *little*, *half* (i.e., 1/2), and *full* are seen in training instructions. For *Handover* and *Fold Shorts*, the dataset only contains 5 demos and 1 demo of the skill, respectively. For *Robot Dog*, it requires delicate operations, as a slight angle when pushing joysticks can make the robot dog deviate.

| TASK NAME | DIMENSION | EXPLANATION |
|---|---|---|
| Wash Cup | Unseen Object ($\mathcal{Q}$1) | To wash one seen and two unseen cups with the faucet |
| Pour Water | Unseen Scene ($\mathcal{Q}$1) | To pour water into the cup in three unseen rooms |
| Pour Water-L-1/3 | Instruction Following ($\mathcal{Q}$2) | To pour water into the cup with the **left hand** until **one-third** full |
| Pour Water-R-2/3 | Instruction Following ($\mathcal{Q}$2) | To pour water into the cup with the **right hand** until **two-thirds** full |
| Handover | 5-Shot Learning ($\mathcal{Q}$3) | To move the marker to the box, where handover is needed due to far distance |
| Fold Shorts | 1-Shot Learning ($\mathcal{Q}$3) | To fold the shorts in half horizontally |
| Robot Dog | Dexterity ($\mathcal{Q}$4) | To push the joystick straight to control the robot dog to walk in a straight line |

(ACT) (Zhao et al., 2023), OpenVLA (Kim et al., 2024), and Octo (Ghosh et al., 2023). ACT is a state-of-the-art method in bimanual manipulation, which uses VAE to model the action distribution. OpenVLA is the largest open-source foundation model (7B), employing the discretization modeling. Octo is a diffusion-based foundation model, and its largest version has only 93M parameters.

**Metric and Hardware.** We employ the success rate as our main metric, which is calculated by dividing successful trials by total trials. *Wash Cup* is tested with 8 trials for each cup (one seen cup, two unseen cups, 24 trials in total). *Pour Water* is tested with 8 trials for each room (three unseen rooms, 24 trials in total). *Pour Water-L-1/3* and *Pour Water-R-2/3* are tested with 8 trials each. *Handover*, *Fold Shorts*, and *Robot Dog* are tested with 25 trials each. All the tests are performed on the ALOHA dual-arm robot (see App. G for hardware configurations). Experimental details, such as the implementation and hyper-parameters, are elaborated in App. H.

**Ablation Study.** Answering $\mathcal{Q}$5, we have conducted ablation studies on the model size, pre-training, and the modeling method to understand their importance. We consider the variants of: *RDT (ours):* the original RDT. *RDT (regress):* RDT without diffusion modeling. It models the deterministic mapping $(\ell, o_t) \mapsto a_t$. *RDT (small):* RDT without large parameters. It has only 166M parameters. *RDT (scratch):* RDT without pre-training. It is trained from scratch during fine-tuning. In Table 2, we evaluate these variants in terms of three dimensions of generalizability. Table 7 provides a comparison of different variants of RDT as well as baselines.

Table 2: **Ablation study results.** Here are the success rates (%) of the original RDT and its three variants in tasks of *Wash Cup* (unseen cup 2, total success rate), *Pour Water* (unseen room 3, total success rate), and *Pour Water-L-1/3* (correct amount sub-task). All the models except *RDT (scratch)* are pre-trained before fine-tuning.

| VARIANT NAME | UNSEEN OBJECT | UNSEEN SCENE | INSTRUCTION FOLLOWING |
|---|---|---|---|
| RDT (regress) | 12.5 | 50 | 12.5 |
| RDT (small) | 37.5 | **62.5** | 25 |
| RDT (scratch) | 0 | 25 | 62.5 |
| RDT (**ours**) | **50** | **62.5** | **100** |

## 5.2 RESULTS ANALYSIS

From the results in Table 3, we can see that RDT consistently outperforms other baselines. This is because RDT employs diffusion with a powerful network architecture to model the distribution of multi-modal actions accurately, while discretization and VAE lack accuracy and expressiveness, respectively. Besides, the large number of parameters after large-scale pre-training provides a lot of prior knowledge, which significantly improves the generalizability. Here is a detailed analysis:

- $\mathcal{Q}$1 & $\mathcal{Q}$2: RDT can zero-shot generalize to unseen objects, scenes, and modalities. In *Wash Cup* and *Pour Water*, RDT can still achieve a high success rate on unseen scenarios, and its performance is not much different from that on seen ones. In contrast, the other baselines cannot even complete the entire task. In *Pour Water-L-1/3* and *Pour Water-R-2/3*, from the third row of Fig. 5 or Fig. 10 (zoomed-in version), we can find that RDT understands precisely which hand to manipulate and how much water to pour and closely follows the instruction through its actions, even though it has never seen words like "one-third" or "two-thirds". It is precisely because of large-scale pre-training that RDT has seen a large number of diverse objects, scenes, and instructions, leading to such strong zero-shot generalization.

Table 3: **Quantitative results.** We report success rates (%) of ACT, OpenVLA, RDT (from scratch, no pre-trained), and RDT (ours, pre-trained) for 7 tasks. Sub-columns in each sub-task cell represent different elements (objects, instructions, scenes). ACT is not language-conditioned and thus unavailable for instruction following. RDT (**ours**) consistently outperforms others.

| Wash Cup: seen cup 1 \| unseen cup 1 \| unseen cup 2 (**Unseen Object**) | | | | | | | | | | | | | | | | | |
| --- | --- | --- | --- | --- | --- | --- | --- | --- | --- | --- | --- | --- | --- | --- | --- | --- | --- |
| | Pick Up Cup | | | Turn On Faucet | | | Get Water | | | Pour Out Water | | | Place Back Cup | | | Total | | |
| ACT | 50 | 12.5 | 37.5 | 0 | 0 | 0 | 0 | 0 | 0 | 0 | 0 | 0 | 37.5 | 0 | 0 | 0 | 0 | 0 |
| OpenVLA | 0 | 0 | 0 | 0 | 0 | 0 | 0 | 0 | 0 | 0 | 0 | 0 | 0 | 0 | 0 | 0 | 0 | 0 |
| Octo | 0 | 0 | 0 | 0 | 0 | 0 | 0 | 0 | 0 | 0 | 0 | 0 | 0 | 0 | 0 | 0 | 0 | 0 |
| RDT (scratch) | 37.5 | 12.5 | 0 | 0 | 12.5 | 12.5 | 0 | 0 | 0 | 37.5 | 12.5 | 0 | 25 | 0 | 0 | 0 | 0 | 0 |
| RDT (**ours**) | 87.5 | 87.5 | 50 | 62.5 | 75 | 50 | 50 | 75 | 50 | 87.5 | 75 | 50 | 87.5 | 62.5 | 50 | **50** | **75** | **50** |

| Pour Water-L-1/3 \| Pour Water-R-2/3 (**Instruction Following**) | | | | | | | | | | | |
| --- | --- | --- | --- | --- | --- | --- | --- | --- | --- | --- | --- |
| | Pick Up Bottle | | Pour Water | | Place Back Bottle | | Total | | Correct Hand | | Correct Amount | |
| OpenVLA | 50 | 0 | 0 | 0 | 0 | 0 | 0 | 0 | 50 | 0 | 0 | 0 |
| Octo | 0 | 0 | 0 | 0 | 0 | 0 | 0 | 0 | 0 | 0 | 0 | 0 |
| RDT (scratch) | 100 | 75 | 75 | 25 | 62.5 | 25 | 62.5 | 25 | 100 | 75 | 62.5 | 12.5 |
| RDT (**ours**) | 100 | 87.5 | 100 | 87.5 | 100 | 87.5 | **100** | **87.5** | **100** | **87.5** | **100** | **75** |

| Pour Water: unseen room 1 \| unseen room 2 \| unseen room 3 (**Unseen Scene**) | | | | | | | | | | | | | | Fold Shorts (**1-Shot**) |
| --- | --- | --- | --- | --- | --- | --- | --- | --- | --- | --- | --- | --- | --- | --- |
| | Pick Up Bottle | | | Pour Water | | | Place Back Bottle | | | Total | | | - | Total |
| ACT | 25 | 87.5 | 25 | 0 | 50 | 12.5 | 0 | 37.5 | 12.5 | 0 | 37.5 | 12.5 | - | 0 |
| OpenVLA | 0 | 0 | 0 | 0 | 0 | 0 | 0 | 0 | 0 | 0 | 0 | 0 | - | 0 |
| Octo | 50 | 0 | 12.5 | 12.5 | 0 | 0 | 12.5 | 0 | 0 | 12.5 | 0 | 0 | - | 4 |
| RDT (scratch) | 62.5 | 100 | 62.5 | 25 | 87.5 | 37.5 | 25 | 75 | 25 | 25 | 75 | 25 | - | 40 |
| RDT (**ours**) | 62.5 | 100 | 62.5 | 62.5 | 100 | 62.5 | 62.5 | 100 | 62.5 | **62.5** | **100** | **62.5** | - | **68** |

| Handover (**5-Shot**) | | | | | | Robot Dog (**Dexterity**) | | | |
| --- | --- | --- | --- | --- | --- | --- | --- | --- | --- |
| | Pick Up Pen | Switch Hand | Drop Pen | Fall into Box | Total | Grab Remote | Push Joystick | Total | Walk Straight |
| ACT | 44 | 0 | 0 | 0 | 0 | 88 | 32 | 32 | 32 |
| OpenVLA | 0 | 0 | 0 | 0 | 0 | 84 | 0 | 0 | 0 |
| Octo | 12 | 0 | 0 | 0 | 0 | 100 | 4 | 4 | 0 |
| RDT (scratch) | 88 | 32 | 24 | 16 | 16 | 100 | 64 | 64 | 32 |
| RDT (**ours**) | 100 | 56 | 56 | 40 | **40** | 100 | 76 | **76** | **48** |

- *Q*3: RDT can learn new skills using only a few shots. In *Handover* and *Fold Shorts*, RDT has learned new and complex skills of handover and folding through few-shot learning, whose action patterns are very different from known skills, while the success rate of others is almost zero. Such improvement is also due to large-scale pre-training. Few-shot learning can help RDT quickly adapt to new working environments, which is of great significance for practical applications.

- *Q*4: RDT can handle dexterous tasks. In *Robot Dog*, RDT accurately controls the angle when pushing the joystick, while others have caused the robot dog to deviate. This is because diffusion, with our powerful network architecture, can model the distribution of multi-modal and nonlinear actions so that the action precision can meet the requirements of dexterous tasks. We also note that the joystick and the remote control are both black, making the joystick not visually apparent. It probably makes ACT prone to failure. In contrast, large-scale pre-training has made RDT learn a better vision-language representation of the joystick concept, improving the recognition capability.

- *Q*5: Large model size, extensive data, and diffusion are all essential factors for our excellence. In Table 2, there is a serious performance drop without any of these factors, demonstrating the necessity of our contributions. In particular, *RDT (scratch)* performs poorly on unseen objects and scenes, indicating that the knowledge from pre-training is critical for generalization.

## 6 CONCLUSION

In this paper, we tackled the challenges of data scarcity and increased manipulation complexity in generalizable bimanual manipulation by developing the Robotics Diffusion Transformer (RDT), a diffusion-based foundation model for language-conditioned visuomotor imitation learning. Our model was pre-trained on an extensive multi-robot dataset and fine-tuned on a self-collected bimanual dataset. We further introduce a Physically Interpretable Unified Action Space to unify action representations across different robots, enhancing robustness and transferability. Outperforming existing methods, RDT not only demonstrates significant improvements in dexterous bimanual capability and instruction following but also achieves remarkable performance in few-shot learning and zero-shot generalization to unseen objects and scenes.

## ACKNOWLEDGMENTS

This work was supported by NSFC Projects (Nos. 92248303, 92370124, 62350080, 62276149, 62061136001, 92270001), BNRist (BNR2022RC01006), Tsinghua Institute for Guo Qiang, and the High Performance Computing Center, Tsinghua University. J. Zhu was also supported by the XPlorer Prize.

## ETHICS STATEMENT

All the data used in this research comes from open-source and well-documented datasets, and we strictly follow all applicable licensing and usage guidelines. Our finetuning dataset is collected by the authors of this paper along with some volunteers.

While RDT is a model trained for scalable, language-conditioned visuomotor policy learning and tested on the ALOHA dual-arm robot, we emphasize that any harmful use of our model is neither intended nor encouraged, and we encourage responsible deployment on real-world robots.

## REPRODUCIBILITY STATEMENT

To reproduce our pre-training and fine-tuning processes, we have provided the code in the repository. We also include instructions for downloading the dataset, how to use the training code, and a guide for deploying on a real machine in the README file. We have fully open-sourced all our code, model weights, and fine-tuning datasets. We refer to the project page for more information.

Please refer to App. D for pre-training dataset details, App. E for fine-tuning dataset details, App. F for RDT training details, App. G for hardware details, and App. H for experimental details and implementation of baselines.

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

## A    ACTION CHUNKING TECHNIQUE

In practice, we find that the errors in action prediction accumulate as the number of historical decisions increases due to the imperfection of the learned policy. This may cause the robot to drift out of the training distribution, reaching hard-to-recover states (Ross et al., 2011). To alleviate this, we prefer to predict multiple actions in one shot, thereby reducing the total number of decisions in a trajectory. In this way, we model $p(\boldsymbol{a}_{t:t+T_a}|\ell, \boldsymbol{o}_t)$, where $\boldsymbol{a}_{t:t+T_a} := (\boldsymbol{a}_t, \ldots, \boldsymbol{a}_{t+T_a-1})$ is an action chunk and $T_a$ denotes the chunk size (Zhao et al., 2023). To adapt Eq. 1 and Eq. 2 to this context, we could simply replace $\boldsymbol{a}_t$ by $\boldsymbol{a}_{t:t+T_a}$. Besides, according to Chi et al. (2023), action chunking is also helpful for improving temporal consistency. It can better consider the coherence of previous and subsequent actions when making decisions and may avoid sudden changes in actions that may cause damage to the robot.

## B    ARCHITECTURE DETAILS

**Encoding of Multi-Modal Inputs.**    Encoding details are outlined below:

- **Low-Dimensional Inputs.** The proprioception $\boldsymbol{z}_t$ and the noisy action chunk $\tilde{\boldsymbol{a}}_{t:t+T_a}$ are first embedded into the unified action space. This space is used to unify the representation of $\boldsymbol{z}_t$ and $\tilde{\boldsymbol{a}}_{t:t+T_a}$ across various robots, which is elaborated in Sec. 4.2. Then, they are encoded into the token space by a shared MLP since they have similar physical meanings. Such continuous encoding can avoid precision loss in contrast to discretized encoding (Brohan et al., 2022; 2023; Kim et al., 2024). For frequency $c$ as well as the diffusion time step $k$, we encode them into the token space through two MLPs, respectively. Afterward, all of them are concatenated together in the length direction to achieve *in-context conditioning* (Peebles & Xie, 2023; Bao et al., 2023), resulting in an input token sequence of length $1 + T_a + 1 + 1$. Finally, position embeddings are added to distinguish different modalities and to inject temporal information in $\tilde{\boldsymbol{a}}_{t:t+T_a}$.
- **Image Inputs.** We encode the RGB images by a frozen SigLIP (Zhai et al., 2023) and utilize an additional MLP to project the output to the token space. To enhance the model's ability to distinguish images based on viewpoint and time steps, we extend traditional sinusoidal positional embeddings to multi-dimensional grids, as shown on the right side of Fig. 3. This modification integrates spatial-temporal information, enabling the model to capture the relationships between input images. Specifically, we adopt the implementation by Liu et al. (2022), employing grid dimensions of $(T_{\text{img}}, N_{\text{cam}}, N_{\text{patch}}, D)$. Here, $N_{\text{cam}}$ represents the number of cameras, set to three in our configuration, and $N_{\text{patch}}$ indicates the number of patches into which each image is divided by the ViT-based Image Encoder and $D$ denotes the embedding dimension.
- **Language Inputs.** Language instruction is encoded by a frozen T5-XXL (Raffel et al., 2020), and an MLP is used to project the output to the token space. When calculating attention for language tokens, we apply the language attention mask to mask out the pad tokens appended during batching.

During training, each input from various modalities is independently masked with a probability of 10%.

**Network Structure of $f_{\boldsymbol{\theta}}$.**    After encoding, we feed the tokens of the low-dimensional inputs into the main network, which is adjusted from Diffusion Transformers (DiTs) with Cross-Attention (Peebles & Xie, 2023) due to their high scalability. For better training stability, we add QKNorm (Henry et al., 2020) into each attention layer and replace each LayerNorm with RMSNorm (Zhang & Sennrich, 2019). In each DiT block's cross-attention layer, we alternately inject language and image tokens rather than simultaneously inject both, avoiding the issue of token imbalance between the two modalities. After $L$ DiT blocks, we normalize the output and project it back to the action space via an MLP decoder.

## C    PHYSICALLY INTERPRETABLE UNIFIED ACTION SPACE

As mentioned in Sec. 4.2, we embed the actions of various robots into one unified space that includes all the main physical quantities of robots. This unified action space has a dimensionality of 128. Table 4 describes each element of the vector in this unified action space. For a specific robot, each

element of the raw action vector is filled into the corresponding position of the unified action vector according to its physical meanings, with the remaining positions being padded.

| Index Range | Element Index | Mapped Physical Quantity |
|---|---|---|
| [0, 10) | 0–9 | Right arm joint positions |
| [10, 15) | 10–14 | Right gripper joint positions |
| [15, 25) | 15–24 | Right arm joint velocities |
| [25, 30) | 25–29 | Right gripper joint velocities |
| [30, 33) | 30–32 | Right end effector positions |
| [33, 39) | 33–38 | Right end effector 6D pose |
| [39, 42) | 39–41 | Right end effector velocities |
| [42, 45) | 42–44 | Right end effector angular velocities |
| [45, 50) | 45–49 | Reserved |
| [50, 60) | 50–59 | Left arm joint positions |
| [60, 65) | 60–64 | Left gripper joint positions |
| [65, 75) | 65–74 | Left arm joint velocities |
| [75, 80) | 75–79 | Left gripper joint velocities |
| [80, 83) | 80–82 | Left end effector positions |
| [83, 89) | 83–88 | Left end effector 6D pose |
| [89, 92) | 89–91 | Left end effector velocities |
| [92, 95) | 92–94 | Left end effector angular velocities |
| [95, 100) | 95–99 | Reserved |
| [100, 102) | 100–101 | Base linear velocities |
| [102, 103) | 102 | Base angular velocities |
| [103, 128) | 103–127 | Reserved |

Table 4: **Description of the unified action space vector.** For single-arm robot cases, its arm is mapped to the "right" arm. For a robot arm with only 6 DoF, its joint positions will be filled in the first 6 of the 10 corresponding positions. The same is true for other physical quantities.

## D  PRE-TRAINING DATASETS

Our pre-training dataset collection includes 46 datasets, with a total scale of 1M+ trajectories and 21TB, making it the largest pre-training collection of robotics datasets to date. Table 5 presents the complete list of our pre-training datasets and their sampling weights. We assign an initial weight of $\sqrt{N_j}$ to each dataset with size $N_j$ and adjust it according to the diversity and quality of each dataset. Compared to linear weighting, this approach prevents excessive sampling of large datasets while ensuring smaller datasets are adequately sampled, thus enhancing the diversity of pre-training samples in each mini-batch. During the pre-training stage, we further observed and adjusted the weights of different datasets based on their intermediate loss results. We increased the weights of those slow-convergent datasets.

**Main Datasets.**    We list some main datasets as follows:

- **RT-1 Dataset** (Brohan et al., 2022) is a large diversve dataset including 130K trajectories with multiple tasks, objects and environments. It is collected across 13 different embodiments, each equipping a single exterior RGB camera. The action space includes the 6D end effector (EEF), gripper open, and base displacement with a control frequency of 3Hz.
- **DROID** (Khazatsky et al., 2024) is a large-scale multi-task dataset with 76K trajectories and 564 scenes. It is collected via teleoperating a Franka Panda 7-DoF Robot Arm, with both wrist and exterior RGB-D cameras. The action space includes 7-DoF joint positions and a gripper width, while the proprioception additionally includes the 6D EEF with a control frequency of 15Hz.
- **RH20T** (Fang et al., 2023) is a comprehensive dataset covering 110K trajectories and 140 tasks. It includes four different robotic embodiments and three different camera views, sampled at a frequency of 10Hz. It also includes both long and short tasks. Its state space is a mix of 6-DoF and 7-DoF joint positions, and it features a third-person perspective RGB-D camera.

- **Mobile ALOHA Dataset** (Fu et al., 2024) is a bimanual dataset containing 1K+ trajectories collected by the Mobile ALOHA robot. Its state space includes base movements and 14-dimensional joint positions of both hands, along with three or four first-person perspective cameras. Some of its data includes wide-ranging perspective changes and base movements, which were originally suitable for imitation learning.
- **Other Datasets.** The other data come from RH20T (Fang et al., 2023), RoboSet (Kumar et al., 2024), BridgeData V2 (Walke et al., 2023), and Open X-Embodiment (Collaboration et al., 2023). Most of them feature different robotic morphology and camera observation, enhancing both heterogeneity and variety of our pretraining datasets.

**Data Cleaning.** Repetitive episodes and episodes of failure are excluded to ensure the quality of the pre-training datasets. We remove blank images, exclude erroneously recorded velocities, and filter out overly short trajectories. Overlength trajectories will be downsampled to avoid unfairness.

**Preprocessing of Multi-Modal Observation/Action Inputs.** We describe the preprocessing details of each modality:

- **Language Instruction $\ell$.** We perform a simple cleaning on the raw text, such as removing illegal characters and extra spaces, capitalizing the beginning of sentences, and adding a period at the end of sentences. We leave the text variable-length.
- **RGB Images $X_{t-T_{\text{img}}+1:t+1}$.** We employ a fixed-length image input strategy. We fix the image input order and format for all robots, with a total of three views: a static exterior view, a right-wrist view, and a left-wrist view, deemed sufficient for the requirements of most bimanual tasks. We treat a single-arm robot's wrist camera as the right-wrist one and pad the unavailable views with the background color. When fed into the model, each image is padded into a square and resized to $384 \times 384$, keeping its origin aspect ratio. Besides, we choose $T_{\text{img}} = 2$ since a history length of two is adequate for most situations, striking a balance between efficiency and performance (Ghosh et al., 2023; Wu et al., 2024). Finally, we can write the image inputs as $X_{t-1:t+1} := (\{X_{t-1}^1, X_{t-1}^2, X_{t-1}^3\}, \{X_t^1, X_t^2, X_t^3\})$.
- **Proprioception $z_t$ and Action Chunk $a_{t:t+T_a}$.** We roughly align the scales of various datasets by unifying the units of physical quantities (m, rad, m/s, rad/s, etc) rather than strictly normalizing to $[-1, 1]$ or $\mathcal{N}(0, 1)$ as in prior work (Chi et al., 2023; Ghosh et al., 2023). For example, "1 (m)" in different datasets corresponds to the same real-world length. Rescaling the physical quantities will destroy such shared properties and thus impair the model's ability to transfer across robots. We also employ the 6D representation (Zhou et al., 2019) for the EEF rotation to overcome the gimbal lock issue.

  Before choosing $T_a = 64$, we have referred to the previous ablation studies by Zhao et al. (2023) and balanced between the performance and computational overhead. Besides, historical proprioceptions $z_i, i < t$ are excluded to prevent the model from learning shortcuts using the low-dimensional inputs only and thus sticking to fixed motion patterns. Instead, we encourage the model to learn generalizable decision-making structures from high-dimensional image features.
- **Control Frequency $c$.** In addressing the challenge posed by differing control frequencies across datasets, we feed the control frequency into the model, allowing the model to take this variation into account when making decisions.

## E  FINE-TUNING DATASET

Our fine-tuning dataset is created using Mobile ALOHA robot (Fu et al., 2024), including 300+ tasks, 6K+ trajectories, and 3M+ frames. It is also one of the largest open-source multi-task bimanual robot datasets to date. Fig. 6 gives a summary of this dataset. We have borrowed 3 tasks (140 episodes in total) from the open-source Songling dataset (Wang et al., 2024).

- **Multi-Modal Features.** We collect the dataset with three RGB cameras positioned at the front and on the left and right grippers. We record dual-arm 6-DoF joint positions and velocities, along with the gripper angles. We manually annotated instructions for each task. To further augment our instructions and align them with the pre-training datasets, we utilize GPT-4-Turbo (Achiam et al., 2023) to generate 100 expanded instructions and one simplified instruction for each task. This multi-modal information further enhances the richness and quality of our dataset.

| Pre-Training Dataset | Sample Percentage (%) |
| --- | --- |
| RT-1 Dataset (Brohan et al., 2022) | 9.00 |
| TACO Dataset (Rosete-Beas et al., 2022) | 1.99 |
| JACO Play Dataset (Dass et al., 2023) | 1.10 |
| Cable Routing Dataset (Luo et al., 2023) | 0.27 |
| NYU Door Opening (Pari et al., 2021) | 0.33 |
| Viola (Zhu et al., 2022a) | 0.40 |
| Berkeley UR5 (Chen et al.) | 1.06 |
| TOTO (Zhou et al., 2023) | 1.06 |
| Kuka (Kalashnikov et al., 2018) | 1.66 |
| Language Table (Lynch et al., 2022) | 3.32 |
| Columbia Cairlab Pusht Real (Chi et al., 2023) | 0.40 |
| Stanford Kuka Multimodal Dataset (Lee et al., 2019) | 1.83 |
| Stanford Hydra Dataset  (Belkhale et al., 2023) | 0.80 |
| Austin Buds Dataset (Zhu et al., 2022b) | 0.23 |
| Maniskill Dataset (Gu et al., 2023) | 5.78 |
| Furniture Bench Dataset (Heo et al., 2023) | 2.36 |
| UCSD Kitchen Dataset (Yan et al., 2023) | 0.40 |
| UCSD Pick And Place Dataset (Feng et al., 2023) | 1.23 |
| Austin Sailor Dataset (Nasiriany et al., 2022) | 0.50 |
| Austin Sirius Dataset (Liu et al., 2023) | 0.80 |
| BC Z (Jang et al., 2021) | 6.91 |
| UTokyo PR2 Opening Fridge (Oh et al., 2023) | 0.30 |
| UTokyo PR2 Tabletop Manipulation (Oh et al., 2023) | 0.50 |
| UTokyo Xarm Pick And Place (Matsushima et al., 2023) | 0.33 |
| UTokyo Xarm Bimanual (Matsushima et al., 2023) | 0.03 |
| Berkeley MVP (Radosavovic et al., 2022) | 0.73 |
| Berkeley RPT (Radosavovic et al., 2022) | 1.00 |
| KAIST Nonprehensile (Kim et al., 2023) | 0.46 |
| Tokyo U LSMO (Osa, 2022) | 0.23 |
| DLR Sara Grid Clamp (Padalkar et al., 2023) | 0.03 |
| Robocook (Shi et al., 2023) | 1.66 |
| Imperialcollege Sawyer Wrist Cam (RethinkRobotics) | 0.43 |
| Iamlab CMU Pickup Insert (Saxena et al., 2023) | 0.83 |
| UTAustin Mutex (Shah et al., 2023b) | 1.29 |
| Fanuc Manipulation (Zhu et al., 2023) | 0.66 |
| Play Fusion (Chen et al., 2023) | 0.80 |
| DROID (Khazatsky et al., 2024) | 10.06 |
| FMB (Luo et al., 2024) | 1.39 |
| Dobb·E (Shafiullah et al., 2023) | 1.20 |
| QUT Dexterous Manipulation (Federico Ceola, 2023) | 0.46 |
| Aloha Dataset (Zhao et al., 2023) | 4.98 |
| Mobile Aloha Dataset (Fu et al., 2024) | 4.98 |
| RoboSet (Kumar et al., 2024) | 4.48 |
| RH20T (Fang et al., 2023) | 10.99 |
| Calvin Dataset (Mees et al., 2022) | 3.32 |
| BridgeData V2 (Walke et al., 2023) | 7.44 |

Table 5: The pre-training datasets and their corresponding weights.

- **Diverse Objects and Scenes.** Our dataset includes diverse tasks and scenes, encompassing more than 300 tasks, including skills such as picking up, inserting, writing, pushing, and pulling. It features 100+ objects with rigid and non-rigid bodies of various sizes and textures. We collect the dataset in 15+ scenes and introduce randomness during data collection for each task, such as varying the initial positions of objects and robots. To further increase diversity, we added random lighting conditions. For instance, pouring water was performed under both normal lighting and changing color conditions. These measures further enhance the diversity of our dataset.

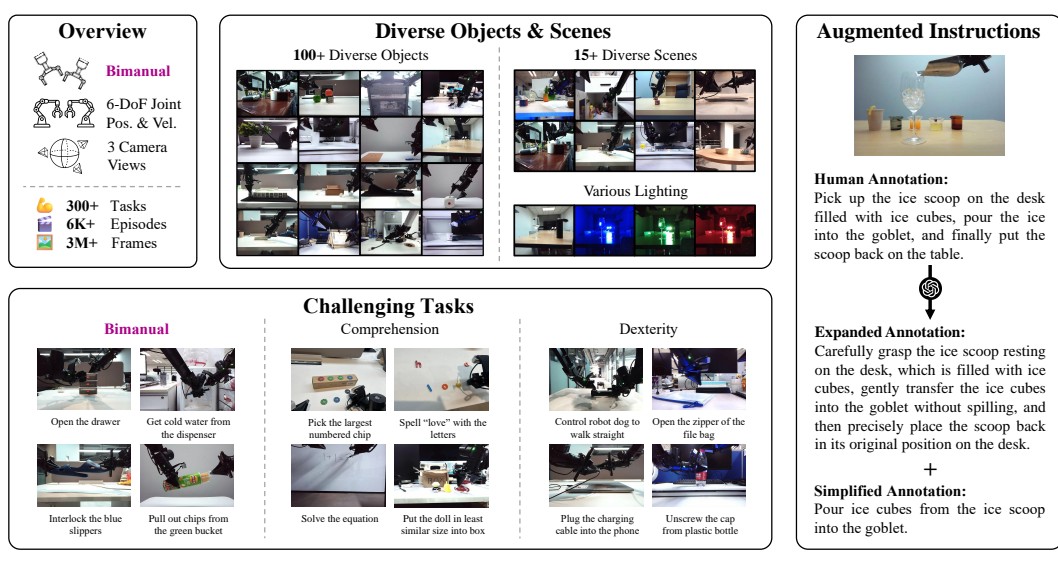

Figure 6: **Fine-Tuning dataset.** Our dataset includes the following key features: (1) **Diverse Objects and Scenes.** Our dataset contains objects with different properties manipulated in different scenes and conditions. (2) **Challenging Tasks.** Our dataset incorporates dexterous manipulation, language and vision comprehension, and bimanual tasks. (3) **Multi-Modal Features.** Our dataset is annotated with rich multi-modal data, including 3-View RGB cameras, joint information, and augmented instructions.

- **Challenging Tasks.** Various challenging tasks are also considered, encompassing dexterous manipulations, such as unscrewing the cap from a plastic bottle, and comprehension tasks, such as spelling "love" with letter blocks. Furthermore, the dataset includes tasks that integrate both dexterity and comprehension, such as solving mathematical equations on the whiteboard. Additionally, our dataset incorporates bimanual tasks, such as inserting the charging cable into the phone. These complex, high-quality tasks further enhance the model's downstream comprehensibility and generalizability.

# F  RDT TRAINING DETAILS

**Platform.** We use Pytorch (Paszke et al., 2019) and DeepSpeed (Rasley et al., 2020) to facilitate parallel training and employ a producer-consumer framework with TensorFlow Dataset (TFD) for fast data loading. Since most of the datasets in the Open X-Embodiment (Collaboration et al., 2023) are stored in the form of `TFRecord`, we convert all pre-training datasets into `TFRecord` for storage. In pre-training, we use the producer process to decompress the data from `TFRecord` and store it in a buffer on the hard disk. At the same time, we use the consumer process to read data from the buffer in a disorderly order and feed it to the model training. This not only decouples the TensorFlow (Abadi et al., 2015) and PyTorch environments but also alleviates the training performance loss caused by the small size of the shuffling buffer in the memory. In the fine-tuning stage, since the dataset is relatively small, we additionally implement a data reading pipeline using the HDF5 dataset for storage.

**Padding Action and Proprioception.** To embed a specific robot action into the 128-dimensional unified action space, we need to pad unavailable action elements. The usual practice is to pad with a 0 value or a specific value. But "0" actually has a physical meaning. For example, a speed of "0" generally represents stillness relative to the ground. This may confuse the model: Does "0" represent stillness or a filler value? To solve this problem, we concatenate the action and proprioception with a 0-1 vector indicating whether each dimension is padded before encoding them into the token space, resulting in a 256-dimensional vector. This can supplement the missing availability information and eliminate confusion.

**Inspecting Training Process.** During training, for every fixed period, we conduct a diffusion sampling and compare the sampled actions with the ground truth of the training dataset. Empirically, we discover a general positive correlation between the Mean Squared Error (MSE) of the two and the performance of deployment on the robot. This observation allows us to monitor the model's training progress easily. When this MSE converges, we can generally stop training. We note that an overly low MSE may also mean overfitting.

**Data Augmentation.** Overfitting is a common challenge in training large neural models, particularly in the fine-tuning phase. We utilize data augmentation techniques to resolve it. We perform image augmentation, including color jittering and image corruption, and add Gaussian noise to the input proprioception with a signal-to-noise ratio (SNR) of 40dB. We also use GPT-4-Turbo to augment and expand the language instructions (Refer to Sec. E for more details on the instruction augmentation).

**Some Fine-Tuning Details.** During fine-tuning, we removed a static part at the beginning of each episode, which might be caused by the operator not reacting after the recording started. Our language instructions are sampled from the original manually annotated instruction, the expanded instructions, and the simplified instruction with a probability of one-third. When the expanded instructions are drawn, we evenly sample one from the 100 expanded instructions corresponding to the task. We did not apply Classifier-Free Guidance (CFG) because we found that this did not improve the performance of the model but instead brought the unstable robot arm behavior.

## G   HARDWARE DETAILS

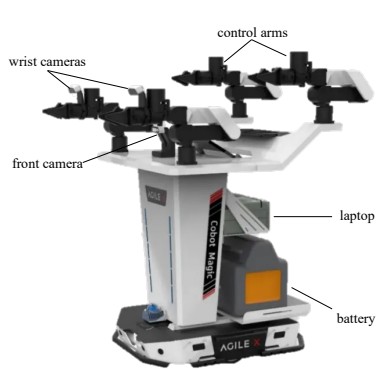

Figure 7: Hardware features.

| Parameter | Value |
|---|---|
| DoF | $7 \times 2 = 14$ |
| Size | $1080 \times 700 \times 1140$ |
| Arm weight | 4.2 kg |
| Arm Payload | 3000 g (peak) |
| | 1500 g (valid) |
| Arm reach | 600 mm |
| Arm repeatability | 1 mm |
| Arm working radius | 653 mm |
| Joint motion range | J1: ±154°, J2: 0°~165° |
| | J3: -175°~0°, J4: ±106° |
| | J5: ±75° , J6: ±100° |
| Gripper range | 0-80 mm |
| Gripper max force | 10 NM |

Table 6: Technical specifications.

We provide a detailed overview of the hardware configuration of our target dual-arm robot. Our model is deployed and evaluated on the Cobot Mobile ALOHA, a robot using the Mobile ALOHA system design (Fu et al., 2024) and manufactured by agilex.ai. The key features of the robot are illustrated in Fig. 7 . It is equipped with two wrist cameras, a front camera, a laptop, and an onboard battery. The robot's technical specifications are listed in Table 6. It is important to note we used the "mobile" ALOHA only to facilitate transportation and testing between various scenes and did not use its autonomous mobility feature during any training or inference stages. Our tasks are still static bimanual manipulation tasks.

## H   EXPERIMENT DETAILS

**Calculation of Total Performance.** The general performance in Fig. 1 of each method is calculated in three steps. Firstly, we calculate the success rate of a method in each task. We take an average of the total success rate and any additional requirement, i.e., the average of the values in the *Total* column and all columns to its right in Table 3. For example, in the *Pour Water-L-1/3*, we take the average of *Total*, *Correct Hand*, and *Correct Amount*. Secondly, we calculate the success rate of

Table 7: **Comparision of different baselines.** We compare baselines as well as different variants of our model in terms of model size, data size, and modeling scheme.

| METHOD NAME | LARGE MODEL | LARGE MULTI-ROBOT DATA | MODELING |
|---|---|---|---|
| ACT (Zhao et al., 2023) | ✗ | ✗ | VAE |
| OpenVLA (Kim et al., 2024) | ✓ | ✓ | Discretization |
| Octo (Ghosh et al., 2023) | ✗ | ✓ | Diffusion |
| RDT (scratch) | ✓ | ✗ | Diffusion |
| RDT (small) | ✗ | ✓ | Diffusion |
| RDT (regress) | ✓ | ✓ | Regression |
| RDT (**ours**) | ✓ | ✓ | Diffusion |

each dimension of *Unseen Object*, *Unseen Scene*, *Instruction Following*, *Few-Shot Learning*, and *Dexterity* by averaging all the tasks in this dimension (see Table 1 for the correspondence). Lastly, we average the success rates of all the dimensions to obtain the overall result.

**Implementation and Hyper-Parameters of RDT.** We list the details of the multi-modal encoders in Table 8 and the model parameter in Table 9. The image history size is $T_{\text{img}} = 2$, the action chunk size is $T_a = 64$, the language token space dimension is 4096, the image token space dimension is 1152, and the token space dimension of RDT is 2048. We use adaptors to align each modality's token dimension to 2048. And all adaptors for multi-modal encoders are with GeLU activation (Hendrycks & Gimpel, 2016).

We use the AdamW optimizer (Adam et al., 2019) with a constant learning rate scheduler and hyper-parameters in Table 10 in the pre-training and fine-tuning stages. The model is pre-trained and finetuned on 48 H100 80GB GPUs for 1M steps and 130K steps, respectively. Due to scheduling reasons, we did not start fine-tuning from the 1M pre-trained checkpoint but chose the 500K checkpoint. During the training stage, we use the DDPM scheduler with a glide cosine scheduler (i.e., `squaredcos_cap_v2`) and a step number of 1000. During the sampling stage, we utilize the DPM-Solver++ (Lu et al., 2022) with a glide cosine scheduler and a sampling step number of 5. During fine-tuning, we also filter out episodes with a length lower than 32 and down-sample those with a length higher than 2048 to 2048.

| Modality | Encoder | Trainable | Adaptor |
|---|---|---|---|
| Language | T5-XXL (Raffel et al., 2020) | N | 2-layers MLP |
| Image | SigLIP (Zhai et al., 2023) | N | 2-layers MLP |
| Action | - | - | 3-layers MLP |

Table 8: Encoder configurations of RDT.

| Model | Layers | Hidden size | Heads | #Params |
|---|---|---|---|---|
| RDT-1B | 28 | 2048 | 32 | 1.2B |

Table 9: Model configurations for RDT.

**Implementation and Hyper-Parameters of ACT.** We directly employed the same architecture and hyper-parameters of ACT as that in the original paper (Fu et al., 2024), except for the hyper-parameters in Table 11. We trained ACT with 90% of the 6K fine-tuning episodes for 8000 epochs (about 8 days in total), while the remaining 10% is treated as the validation set. We took the checkpoint at epoch 5413 as the final outcome, according to the best performance in the validation set.

**Implementation and Hyper-Parameters of OpenVLA.** We adopt the official implementation (https://github.com/openvla/openvla) and flagship pre-trained model and checkpoint

| Hyper-Parameter | Value |
|---|---|
| Batch Size | $32\times48$ |
| Learning Rate | $1 \times 10^{-4}$ |
| Mixed Precision | `bf16` |
| Warm-Up Steps | 500 |
| $\beta_1$ | 0.9 |
| $\beta_2$ | 0.999 |
| Weight Decay | $1 \times 10^{-2}$ |
| $\epsilon$ | $1 \times 10^{-8}$ |

Table 10: Hyper-parameters for both pre-training and fine-tuning RDT.

| Hyper-Parameter | Value |
|---|---|
| Batch Size | $80\times4$ |
| Learning Rate | $9 \times 10^{-5}$ |
| Learning Rate for Backbone | $4 \times 10^{-5}$ |

Table 11: Adapetd hyper-parameters of ACT.

at `https://huggingface.co/openvla/openvla-7b`. For each task in evaluation, we further fine-tune the officially pre-trained OpenVLA with all the task-relevant demonstrations ($\sim 100$ episodes) from the fine-tuning dataset to facilitate convergence and train the model to around $95\%$ action token accuracy as suggested by Kim et al. (2024) (`https://github.com/openvla/openvla/issues/12#issuecomment-2203772810`). Additionally, we experimented with both full-parameter tuning and LoRA methods using the entire dataset but did not achieve sufficient action token accuracy (approximately $60\%$) for deployment upon convergence (see Fig. 8). According to real-robot testing, such non-convergent checkpoints exhibit completely static or random behaviors in the deployment.

Concretely, we adhere to the same hyper-parameters claimed in Kim et al. (2024) for fine-tuning via LoRA (Hu et al., 2021) as detailed in Table 12.

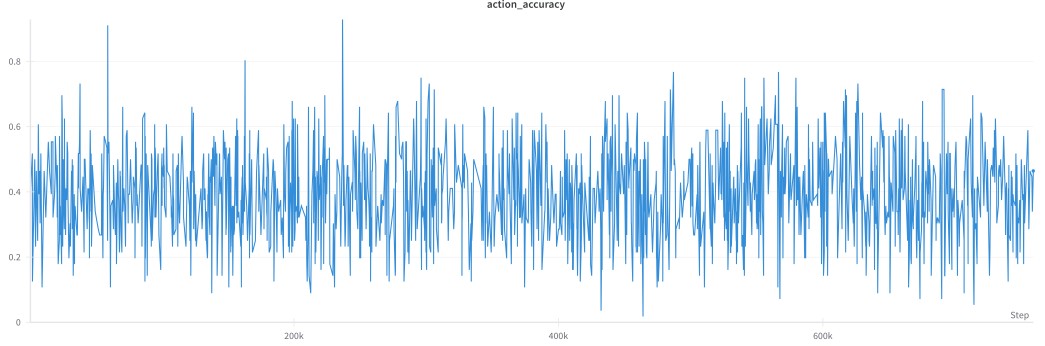

Figure 8: The accuracy of action token prediction fluctuates rather than converges with the number of training steps when fine-tuning OpenVLA with the full fine-tuning dataset.

**Implementation and Hyper-Parameters of Octo.** We utilize the official implementation available at `https://github.com/octo-models/octo` and the most comprehensive pre-trained model, `octo-base-1.5`, hosted at `https://huggingface.co/rail-berkeley/octo-base-1.5`. We follow the officially recommended practices for fine-tuning a bimanual robot, detailed in `https://github.com/octo-models/octo/blob/main/examples/02_finetune_new_observation_action.py`, employing a full-parameter approach. Additionally, we have incorporated an extra image tokenizer to process images from the right-wrist camera,

| Hyper-Parameter | Value |
|---|---|
| Batch Size | 16×8 |
| Learning Rate | $2 \times 10^{-5}$ |
| Lora Rank | 32 |
| Image Augmentation | True |

Table 12: Hyper-parameters of fine-tuning OpenVLA for bimanual manipulations.

enhancing the system's manipulation capabilities. Furthermore, by integrating image augmentation during the fine-tuning process, we enhance the performance upon deployment in real-world robotics. We replicate the wrist image tokenizer from the pre-trained model to initialize the right-wrist image tokenizer. Similar to OpenVLA, we only fine-tune octo with the task-relevant demonstrations for each evaluation tasks, for we do not observe sufficient test MSE (approximately $10^{-1}$) for deployment upon convergence (Fig. 9). Concretely, we apply the default hyper-parameters with variations listed in Table 13:

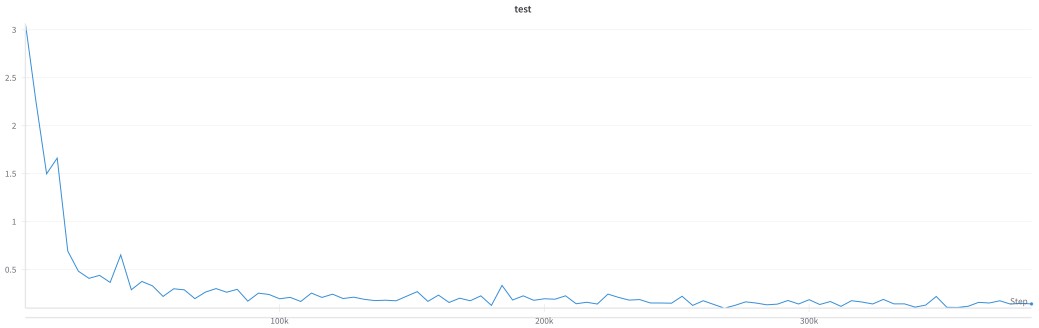

Figure 9: The test MSE of action prediction fluctuates rather than converges with the number of training steps when fine-tuning Octo with the full fine-tuning dataset.

oct

| Hyper-Parameter | Value |
|---|---|
| Action Head Type | DiffusionActionHead |
| Batch Size | 8×8 |
| Action Chunk Size | 8 |
| Image Augmentation | RandomBrightness(0.1) RandomContrast(0.9, 1.1) RandomSaturation(0.9, 1.1) RandomHue(0.05) |

Table 13: Hyper-parameters of fine-tuning Octo for bimanual manipulations.

# I   MORE RESULTS

We further provide a zoom-in view for water-level across 8 trails in instruction-following evaluation in Fig. 10.

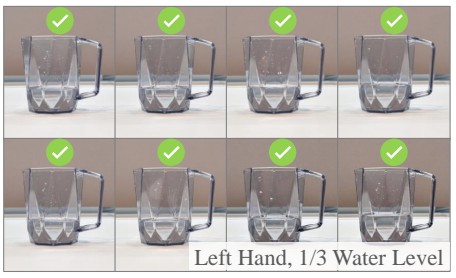 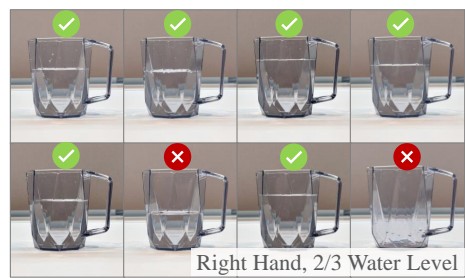

Figure 10: Visulization of the resulting water levels across 8 trails in *Pour Water-L-1/3* and *Pour Water-R-2/3*. **Left:** The water level completed by RDT in each trial is extremely close to the ground-truth 1/3 standard. **Right:** RDT made one mistake in pouring (empty cup) and one mistake in water level, but the other trials were in roughly good agreement with 2/3.

