# OpenReview forum: "RDT-1B: a Diffusion Foundation Model for Bimanual Manipulation"
_ICLR.cc/2025/Conference — ICLR 2025 Poster_

### Official Review · Reviewer_dG58 · 2024-10-29

**Soundness:** 2
**Presentation:** 3
**Contribution:** 2
**Rating:** 6
**Confidence:** 3

**Summary:**

This paper proposes a unified action representation to align different robots, facilitating pre-training on diverse robot datasets for bimanual manipulation. Additionally, it introduces a diffusion transformer-based architecture with several modifications for enhancing policy learning, and scaling up with large datasets.

**Strengths:**

A foundational model for bimanual manipulation is absent in the current community, which is an important direction; aligning different robot embodiments is also a crucial question for pre-training on large-scale datasets. The proposed action representation is simple yet effective for pre-training on diverse robot datasets.

**Weaknesses:**

Compared to the robot datasets used for pre-training the baseline, such as OpenVLA, this paper appears to use a more diverse set of datasets, including additional bimanual manipulation datasets like ALOHA and Mobile ALOHA, contributing nearly 10% of the total datasets. Fine-tuning a baseline pre-trained on single-arm datasets for a bimanual manipulation setting may result in poor performance on bimanual tasks, making it difficult to demonstrate that using the diffusion transformer architecture is superior to using a large language model as the pre-training backbone.

**Questions:**

1. "Does the RDT-1B fine-tuning use the entire self-collected dataset, and is it not fine-tuned separately for each task in the evaluation?"
2. "Regarding the computation of success rate for the 'wash cup' task with 'seen cup1': the success rate (SR) for 'get water' is 50, for 'pour water' is 87.5, and for 'place back cup' is also 87.5, yet the overall SR is listed as 50. Since the 'get water' subtask has an SR of 50, and the following subtasks have SRs below 100, how is the total SR calculated as 50?"

---

> ### Author Response · Authors · 2024-11-18
> **Thank you for the valuable review (1/3)**
>
> We sincerely thank you for your insightful review of our submission. Your feedback is invaluable, and we appreciate the time and effort you dedicated to evaluating our work.
>
> In response, we have conducted additional experiments to strengthen our evaluation and address your concerns. Below, we provide the results and clarifications to resolve any confusion.
>
> ## Q1: Compared to the robot datasets used for pre-training the baseline, such as OpenVLA, this paper appears to use a more diverse set of datasets, including additional bimanual manipulation datasets like ALOHA and Mobile ALOHA, contributing nearly 10% of the total datasets.
>
> Thank you for pointing it out. We have now extended the pre-training of the officially released models (Octo and OpenVLA) using the largest open-source bi-manual manipulation datasets, including ALOHA (static and mobile) and our fine-tuning dataset, totaling over 7,000 episodes. However, even after this additional pre-training, the models still failed to converge to a deployable loss level, resulting in random or meaningless patterns of movement in real-world scenarios  (such phenomenon is elaborated in App. H). The detailed results are depicted in Table a.
>
> *Table a: The performance comparison of different methods further pre-trained on bimanual dataset (ALOHA Static and ALOHA Mobile and our finetune dataset). They are further finetuned with whole fine-tuning dataset.*
>
> *Wash Cup: seen cup 1 | unseen cup 1 | unseen cup 2 (**Unseen Object**)*
>
> |          | Pick Up Cup |      |      | Turn On Faucet |      |      | Get Water |      |      | Pour Out Water |      |      | Place Back Cup |      |      | Total |      |      |
> | -------- | ----------- | ---- | ---- | -------------- | ---- | ---- | --------- | ---- | ---- | -------------- | ---- | ---- | -------------- | ---- | ---- | ----- | ---- | ---- |
> | Octo     | 0           | 0    | 0    | 0              | 0    | 0    | 0         | 0    | 0    | 0              | 0    | 0    | 0              | 0    | 0    | 0     | 0    | 0    |
> | OpenVLA | 0           | 0    | 0    | 0              | 0    | 0    | 0         | 0    | 0    | 0              | 0    | 0    | 0              | 0    | 0    | 0     | 0    | 0    |
>
>  *Pour Water-L-1/3 | Pour Water-R-2/3 (**Instruction Following**)*
>
> |         | Pick Up Bottle |      | Pour Water |      | Place Back Bottle |      | Total |      | Correct Hand |      | Correct Amount |      |
> | ------- | -------------- | ---- | ---------- | ---- | ----------------- | ---- | ----- | ---- | ------------ | ---- | -------------- | ---- |
> | Octo    | 0              | 0    | 0          | 0    | 0                 | 0    | 0     | 0    | 0            | 0    | 0              | 0    |
> | OpenVLA | 50             | 0    | 0          | 0    | 0                 | 0    | 0     | 0    | 50           | 0    | 0              | 0    |
>
>  *Pour Water: unseen room 1 | unseen room 2 | unseen room 3 (**Unseen Scene**)*
>
> |         | Pick Up Bottle |      |      | Pour Water |      |      | Place Back Bottle |      |      | Total |      |      |
> | ------- | -------------- | ---- | ---- | ---------- | ---- | ---- | ----------------- | ---- | ---- | ----- | ---- | ---- |
> | Octo    | 0              | 0    | 0    | 0          | 0    | 0    | 0                 | 0    | 0    | 0     | 0    | 0    |
> | OpenVLA | 0              | 0    | 0    | 0          | 0    | 0    | 0                 | 0    | 0    | 0     | 0    | 0    |
>
>  *Fold Shorts (**1-Shot**)*
>
> |         | Total |
> | ------- | ----- |
> | Octo    | 0     |
> | OpenVLA | 0     |
>
>  *Handover (**5-Shot**)*
>
>
> |         | Pick Up Pen | Switch Hand | Drop Pen | Fall into Box | Total |
> | ------- | ----------- | ----------- | -------- | ------------- | ----- |
> | Octo    | 0           | 0           | 0        | 0             | 0     |
> | OpenVLA | 0           | 0           | 0        | 0             | 0     |
>
> In summary, despite extensive efforts in re-implementing and fine-tuning these models for optimal performance, the issue of insufficient convergence persists. For OpenVLA, there might be a broader scientific challenge in scaling vision-language-action (VLA) models effectively for **bi-manual manipulation**, which could be a possible reason why they do not consider bi-manual manipulation [*1]. For Octo, other researchers have also reported the phenomenon of non-convergence and random behaviors of the robot when adapting Octo to bi-manual manipulation [*2]. We can conclude that bimanual manipulation is challenging for previous foundation models, which is addressed by designing a unified action space and a powerful architecture in our RDT. While beyond the scope of this work, we believe adapting VLA to bimanual manipulation will be a promising direction.
>
> [*1] https://github.com/openvla/openvla/issues/48
>
> [*2] https://github.com/octo-models/octo/issues/29

---

> > ### Comment · Reviewer_dG58 · 2024-11-25
> >
> > can you illustrate more about how you modify and pre-train the baseline on bimanual datasets since the original baseline is for single-arm, and how long it takes for pertaining and fine-tuning the baselines for bi-manual datasets?

---

> ### Author Response · Authors · 2024-11-18
> **Thank you for the valuable review (2/3)**
>
> ## Q2: Does the RDT-1B fine-tuning use the entire self-collected dataset, and is it not fine-tuned separately for each task in the evaluation?
>
> By default, all baseline models are fine-tuned using the entire self-collected dataset and tested by a multi-task setting (one model for all tasks). However, due to the limited model capacity of Octo and OpenVLA, they experience issues with insufficient convergence when fine-tuned this way (more details are elaborated in **Q1** and App. H). As an alternative, we fine-tuned these models separately for each task to enhance their performance and tested each task with the model fine-tuned on the data of this task (one model for one task).
>
> To facilitate a fair comparison, we also fine-tune RDT on one of the tasks. From the results above, we can find that RDT performs even better than the original, which means that the multi-task setting is much more challenging than the single-task setting. For the former, the model needs to have sufficient capacity to simultaneously fit the distribution of multiple tasks. So, it is fair to compare the performance of the multi-task RDT with that of the single-task versions of baselines.
>
>
>  *Pour Water: unseen room 1 (**Unseen Scene**)*
>
>
> |                            | Pick Up Bottle | Pour Water | Place Back Bottle | Total |
> | -------------------------- | -------------- | ---------- | ----------------- | ----- |
> | ACT                        | 25             | 0          | 0                 | 0     |
> | OpenVLA                    | 0              | 0          | 0                 | 0     |
> | Octo                       | 50             | 12.5       | 12.5              | 12.5  |
> | RDT (original, multi-task) | 62.5           | 62.5       | 62.5              | 62.5  |
> | RDT (single-task)          | 100            | 87.5       | 87.5              | 87.5  |

---

> ### Author Response · Authors · 2024-11-18
> **Thank you for the valuable review (3/3)**
>
> ## Q3: Regarding the computation of success rate for the 'wash cup' task with 'seen cup1': the success rate (SR) for 'get water' is 50, for 'pour water' is 87.5, and for 'place back cup' is also 87.5, yet the overall SR is listed as 50. Since the 'get water' subtask has an SR of 50, and the following subtasks have SRs below 100, how is the total SR calculated as 50?
>
> We thank the reviewer for bringing up this question, which helped us improve our presentation. The success rate (SR) of a subtask here refers to the percentage: $P(\text{completing this subtask})$, not the percentage: $P(\text{completing this subtask}\mid \text{completing all subtasks before this subtask})$. There are also scenarios that a subtask is completed without completing all previous subtasks. In the particular example you mentioned, the 87.5% SR of "pour out water" means that in 87.5% of cases, the agent completed the action of "pour out water", 50% of which there was enough water in the cup ("get water" is successful), and 37.5% of which there were only a few drops of water in the cup ("get water" is failed). The 50% total SR means that in 50% of cases, the agent completed all the subtasks.

---

> ### Author Response · Authors · 2024-11-23
> **Look forward to further feedback**
>
> Dear Reviewer dG58,
>
> We would like to express our sincere gratitude for your insightful feedback. We hope our response and additional experiments can meet your expectations and await any further comments you may have.
>
> Best,
> Authors

---

> ### Author Response · Authors · 2024-11-25
> **Thank you for your thoughtful question**
>
> Thank you for your thoughtful question. Please find our detailed response below:
>
>
>
> **1. Modifications for Bimanual Datasets**
>
>
>
> - **Input Modifications**:
>
>     The original models were designed for robots using a third-person view camera, which provides a comprehensive view of the manipulation scene. In contrast, ALOHA employs a single first-person camera and two wrist cameras. Using only one camera can result in unexpected information loss and partial observability. Therefore, we modified the input to include three images: one from the front-facing camera and two from the wrist cameras, instead of relying solely on a third-person view
>
> - **Output Modifications**:
>
>     The original models predict the 7DOF end-effector pose for control, while ALOHA utilizes joint positions for control. Consequently, we adapted the models to predict target joint positions instead of end-effector poses.
>
>
>
> **2. Training and Fine-Tuning Details for OpenVLA**
>
>
>
> - **Pretraining**: We continued pretraining OpenVLA on an integrated bimanual dataset using 2x8 H100 GPUs with a global batch size of 64 for 300k steps. This process took approximately **3 days**.
> - **Fine-tuning**:
> For fine-tuning, we used the same resources, training the model for 50k steps, which took approximately **13 hours**. The model achieved an average classification accuracy of **30%**.
>
>
>
> **3. Training and Fine-Tuning Details for Octo**
>
>
>
> - **Pretraining**: Octo was pretrained on the bimanual dataset using a single 8x H100 setup with a batch size of 32 for 300k steps.
> - **Fine-tuning**:
> For fine-tuning, we used the same setup (1x8 H100 with a batch size of 32) for 700k steps. The model achieved a validation set loss of approximately **5e-2**.
>
>
>
> We hope this clarifies our approach. Please feel free to reach out if you have any further questions or require additional details!

---

> > ### Comment · Reviewer_dG58 · 2024-11-26
> >
> > To be honest, I still have concerns about the comparison with the baseline. The baseline is not pretrained from scratch on the exact same dataset but is first pretrained on a single-arm dataset, followed by modifications to the input and output for bimanual tasks, and then further pretraining. This process should impact the baseline's performance. I would have preferred to see the baseline adopt the same representation proposed in this paper, pretrained from scratch, and then fine-tuned. From my perspective, this approach would more convincingly demonstrate the advantages of the proposed diffusion backbone compared to the adapted VLA. Nevertheless, this work represents a foundational model for bimanual manipulation, capable of handling diverse tasks without relying solely on fine-tuning with task-specific demonstrations. As such, I am inclined to raise my score to a weak accept, though I still have reservations regarding the backbone comparison.

---

> > > ### Author Response · Authors · 2024-12-04
> > >
> > > Thank you for raising this concern and for recognizing the value of our work. We respectfully disagree with the suggestion that the baseline should adopt our unified action space and be pretrained from scratch. The unified action space is a core contribution of our method, enabling effective utilization of both single-arm and dual-arm datasets to enhance bimanual performance. Adapting this key innovation to the baseline would blur the distinction between our approach and the baseline, as it is precisely the lack of a robust unified representation that limits other methods from leveraging large and diverse datasets effectively. Additionally, pretraining a baseline model from scratch using our representation would require substantial computational resources, which is beyond the practical scope of this work. Instead, to address your concern about fairness, we included dual-arm data from our pretraining dataset when fine-tuning the baseline, ensuring the most equitable comparison possible within reasonable constraints.
> > >
> > > Furthermore, we agree that isolating the backbone’s effect is important for understanding its contribution. To this end, we performed additional experiments focusing on single-arm tasks without altering the action spaces of OpenVLA and Octo, as their pretrained datasets are comparable in scale to ours in single-arm tasks. This setup meets the fairness criterion for comparing backbones while isolating other factors. The results (see the next response) clearly demonstrate that even under these conditions, our diffusion backbone exhibits significant advantages, reaffirming its strength independent of the unified action space. We hope this clarifies our approach and rationale and appreciate your thoughtful critique, which has motivated these further analyses.

---

> > > ### Author Response · Authors · 2024-12-04
> > > **Addtional Existing Simulation Benchmark of A Novel Single-Arm Franka Hardware**
> > >
> > > To further demonstrate the effectiveness of RDT, we conducted comparative experiments with other baseline methods using the ManiSkill simulation benchmark (an existing benchmark). Due to time and resource constraints, we selected the PickCube-v1 task for our experiments. The objective of PickCube-v1 is to grasp a red cube and move it to a specified target position. Detailed information about this task can be found in the [official ManiSkill documentation](https://maniskill.readthedocs.io/en/latest/tasks/table_top_gripper/index.html#pickcube-v1).
> > > The simulator utilizes a single-arm Franka Panda robot, which differs from our current hardware setup. This variation allows us to assess the effectiveness of RDT across different hardware configurations, thereby demonstrating its versatility.
> > >
> > > ## Training Details
> > >
> > > ### Data
> > >
> > > Utilizing the [official ManiSkill repository](https://github.com/haosulab/ManiSkill), we generated 1,000 trajectories through motion planning. The initial action mode of these trajectories is absolute joint position control and we subsequently converted them into **delta end-effector pose** control to align with the pre-training action space of OpenVLA and Octo. We strictly adhered to the official codebases of OpenVLA and Octo, modifying only the dataset-loading scripts. Consequently, we finetuned OpenVLA and Octo using the delta end-effector pose data. For RDT and Diffusion-Policy we leverage joint position control data for training which is aligned with our pre-training stage as well.
> > >
> > > ### Training
> > >
> > > - OpenVLA is fine-tuned from the officially released pre-trained checkpoint with LoRA-rank 32 until converge, achieving a final training accuracy of 95%, which has reach the [success rate requirement](https://github.com/openvla/openvla) suggested by the authors. The loss curve and accuracy curve are at https://anonymous.4open.science/r/rdt-sim-eval-118C/openvla/openvla-training-acc.png
> > > - Octo is fine-tuned from the officially released pre-trained checkpoint for 450k iterations until converge. The loss curve is at https://anonymous.4open.science/r/rdt-sim-eval-118C/octo/octo-training-loss.png
> > > - Diffusion-Policy is trained from scratch for 1000 epochs. We select the checkpoint of 700 epoch which has the lowest validation sample loss of 1e-3.
> > > - RDT is fine-tuned from our released pre-trained checkpoint for 150k iterations.
> > >
> > > ### Results
> > >
> > > Each method is evaluated over 250 trials (10 random seeds with 25 trials per seed). The quantitative results, including success rate mean and std value across 10 random seeds are presented below:
> > >
> > >
> > > |              | RDT   | OpenVLA | Octo | Diffusion-Policy |
> > > | ------------ | ----- | ------- | ---- | ---------------- |
> > > | Success Mean | 72.4% | 20%     | 0%   | 44%              |
> > > | Success Std  | ±2.8% | 0%      | 0%   | 0%               |

---

### Official Review · Reviewer_kiFc · 2024-11-04

**Soundness:** 4
**Presentation:** 4
**Contribution:** 4
**Rating:** 8
**Confidence:** 3

**Summary:**

This paper presents the Robotics Diffusion Transformer (RDT), a large-scale diffusion-based model for bimanual robotics. RDT introduces a Physically Interpretable Unified Action Space to standardize actions across robots, enhancing generalization, and uses diffusion models to handle complex, continuous action distributions. Pre-trained on the largest multi-robot dataset and fine-tuned on a custom bimanual dataset, RDT demonstrates strong zero-shot and few-shot generalization, outperforming existing methods in dexterous, real-world tasks.

**Strengths:**

- The paper introduces a novel application of diffusion models to bimanual manipulation, addressing the high-dimensional, multi-modal action space through a Physically Interpretable Unified Action Space. This approach is a creative extension of diffusion models in robotics, particularly for dual-arm coordination, a challenging domain with limited prior work.

- The model is rigorously tested, with comprehensive experiments demonstrating superior performance over existing baselines. The use of the largest multi-robot dataset and a specialized bimanual dataset for fine-tuning enhances the validity of results and supports the model's effectiveness.

- This work contributes substantially to the field by advancing foundation models for robotic manipulation. RDT’s capabilities for zero-shot generalization, few-shot learning, and instruction following mark a significant step towards adaptable and scalable robotic models, with promising implications for real-world applications.

**Weaknesses:**

- The paper introduces a Physically Interpretable Unified Action Space for handling data heterogeneity, but additional details on potential limitations or failure cases during training with highly diverse data would be beneficial. This could include examples where action standardization might lead to loss of unique features across robots.

- Although the experiments show impressive results, expanding the evaluation to more varied and complex real-world tasks (beyond the 6,000-episode dataset) and more hardwares(beyond ALOHA) could further validate RDT's robustness.

- The paper proposes several innovative multi-modal encodings (e.g., masking, cross-attention) but lacks ablation studies on these design choices. Showing how each component contributes to performance could clarify their impact on handling visual and language-conditioned tasks effectively.

- There's a typo at L76, it should be "data" instead of "date" .

**Questions:**

- Could the authors elaborate on why diffusion models were specifically chosen over other generative methods like VAEs or GANs for this task? While diffusion models show high expressiveness, a comparison or rationale would clarify their unique benefits in bimanual manipulation.

- The Physically Interpretable Unified Action Space is innovative, but how does it handle robots with vastly different kinematics or action constraints?

---

> ### Author Response · Authors · 2024-11-18
> **Thank you for the valuable review (1/3)**
>
> We sincerely thank you for your detailed and constructive feedback, which has greatly helped us improve our work. We appreciate your recognition of the novelty and practical value of our approach, as well as the thoroughness of our experiments.
>
> In response to your comments, we have carefully addressed each point you mentioned. We hope our revisions and additional experiments meet your expectations. Below, we provide detailed responses to each of your points.
>
> ## Q1: This could include examples where action standardization might lead to the loss of unique features across robots.
>
> Thanks for your advice. The most direct example is the situation we encounter in our pre-training dataset. Our collection contains datasets of multiple robots, each of which is small in size and often contains limited types of tasks. If we standardize each dataset with its mean and variance, there may be a relatively large variance, which is not conducive to model learning. Moreover, independent standardization will cause the action space of each robot to be different, thereby destroying the alignment across robots, and making it difficult for the model to learn shared physical laws. In general, to align different action spaces, we need pairwise data: the trajectories of different robots performing the same task in the same scene. Such data is extremely difficult to obtain. Therefore, we believe that separate standardization is harmful.

---

> ### Author Response · Authors · 2024-11-18
> **Thank you for the valuable review (2/3)**
>
> ## Q2: Expanding the evaluation to more varied and complex real-world tasks and more hardware could further validate RDT's robustness.
>
> Thank you for your thoughtful feedback and insightful suggestions. In response to your recommendation, we have expanded our evaluation to include a broader range of real-world tasks that are both diverse and more challenging. Despite constraints in time and resources, we designed four additional experiments to assess three key dimensions: **Unseen Scene**, **Unseen Object**, and **Instruction Following**.
>
> ### Task Definition
>
> Below, we provide an overview of each task.
>
> | **TASK NAME**                   | **DIMENSION**         | **EXPLANATION**                                              |
> | ------------------------------- | --------------------- | ------------------------------------------------------------ |
> | Stack Tomato Can (STC)          | Unseen Scene          | To stack the right tomato can onto the left one (in seen and unseen scenes) |
> | Stack Tomato Can Mahjong (STCM) | Unseen Object         | To stack the tomato can onto the (unseen) foam mahjong       |
> | Put Orange Lunchbox (POL)       | Instruction Following | To pick up the orange and place it into the paper lunchbox (where the robot needs to distinguish between orange and apple) |
> | Put Apple Lunchbox (PAL)        | Instruction Following | To pick up the green apple and place it into the paper lunchbox |
>
> Task visualizations can be found at https://anonymous.4open.science/r/rdt-demo-543D/addtional_experiment_visualization.png
>
> ### Implementation Details
>
> The implementation details for each method are described as follows:
>
> **1. Training Details for ACT**
>
> ACT was fully trained using our finetune dataset. Since ACT does not support language instructions as input, it was excluded from experiments involving **Instruction Following** tasks.
>
> **2. Training and Fine-Tuning Details for OpenVLA**
>
> - **Pretraining**: We continued pretraining OpenVLA on an integrated bimanual dataset (a combination of Mobile ALOHA dataset and our finetune dataset).
> - **Fine-tuning**: We fine-tuned OpenVLA on the entire fine-tuning dataset. The model achieved an average classification accuracy of **30%**.
>
>
> **3. Training and Fine-Tuning Details for Octo**
>
> - **Pretraining**: Octo was pre-trained on the same bimanual dataset as OpenVLA.
> - **Fine-tuning**: We fine-tuned Octo on the entire fine-tuning dataset. The model achieved a validation set loss of approximately **5e-2**.
>
> **4. Implementation of RDT**
>
> For RDT, we utilized our previously fine-tuned checkpoints in all experiments, consistent with prior evaluations.
>
> ### Experiment Results
>
> Each task was evaluated over **8 trials**. The quantitative results are shown below:
>
> |               | **STC (Seen)** |      | **STC (Unseen 1)** |      | **STC (Unseen 2)** |      | **STCM** |      | **POL** |      | **PAL** |      |
> | ------------- | -------------- | ---- | ------------------ | ---- | ------------------ | ---- | -------- | ---- | ------- | ---- | ------- | ---- |
> | **ACT**       | 62.5           | 37.5 | 0                  | 0    | 0                  | 0    | 50       | 0    | -       | -    | -       | -    |
> | **OpenVLA**   | 12.5           | 0    | 0                  | 0    | 0                  | 0    | 0        | 0    | 0       | 0    | 0       | 0    |
> | **Octo**      | 0              | 0    | 0                  | 0    | 0                  | 0    | 0        | 0    | 0       | 0    | 0       | 0    |
> | **RDT(Ours)** | 75             | 50   | 75                 | 37.5 | 62.5               | 25   | 75       | 50   | 87.5    | 87.5 | 75      | 75   |
> ### Results Analysis
>
> We highlight the challenges of each task and evaluate method performance. **Stack Tomato Can (STC)** involves adapting to new environments and precise placement of the cylindrical, slippery tomato can. **Stack Tomato Can Mahjong (STCM)** requires generalization to unseen foam mahjong pieces.  **Put Orange Lunchbox (POL)** and **Put Apple Lunchbox (PAL)** demand instruction-following to distinguish and pick the correct object.
>
> In the seen scene, ACT can grasp and occasionally place the tomato can but fails in the unseen scene, showing erratic movements. Octo reliably reaches targets but fails to grasp due to ineffective gripper closure. OpenVLA can grasp in the seen scene but struggles with placement and frequently misidentifies objects, while also failing to generalize to unseen scenes. In contrast, RDT demonstrates robust generalization, successfully handling unseen objects and scenes while following instructions accurately.
>
> ### Including Additional Hardware
>
> We are actively working on it; however, due to shipping delays, there is a possibility that it may not arrive in time for further evaluations. We will incorporate these results into future updates as resources become available.

---

> ### Author Response · Authors · 2024-11-18
> **Thank you for the valuable review (3/3)**
>
> ## Q3: Showing how each component contributes to performance could clarify their impact on handling visual and language-conditioned tasks effectively.
>
> Yes, we completely agree with you that more ablation studies will help to deepen our understanding of the model. Since pre-training requires a lot of GPUs and time and real-robot experiments also require effort (dozens of trials per task), with limited resources, we decided to perform ablation studies on our key architectural innovations (e.g., improved normalization, MLP decoder, alternating condition injection) to highlight the necessity of our contribution. In future work, we will conduct ablation experiments on more design choices once additional resources are permitted.
>
> ## Q4: There's a typo at L76, it should be "data" instead of "date".
>
> Thanks for your careful reminder. We have corrected it in the revised manuscript.
>
> ## Q5: Could the authors elaborate on why diffusion models were specifically chosen over other generative methods like VAEs or GANs for this task?
>
> This is a very good question to gain deeper insights into our probabilistic modeling choice. VAEs assume that the data are related to a Gaussian latent variable and sample the data by Gaussian distribution with learnable mappings. Such assumptions can reduce sampling quality [*1], which is critical for dexterous tasks demanding high action precision.
>
> GANs involve adversarial training between a generator and a discriminator. This setup needs carefully tuned loss functions to avoid problems like mode collapse, where the generator produces limited diversity. Adversarial training is generally unstable, which requires specialized stability techniques like gradient penalty and spectral normalization [*2].
>
> In contrast, diffusion models excel in both sampling quality and training stability. Usually, its drawback is the sampling speed, which is minor in our settings since actions have a much lower dimension than images and videos. Therefore, diffusion models become an ideal choice for robot learning.
>
>
> [*1] Dai, B., \& Wipf, D. (2019). Diagnosing and enhancing VAE models. arXiv preprint arXiv:1903.05789.
>
> [*2] https://www.sapien.io/blog/gans-vs-diffusion-models-a-comparative-analysis
>
> ## Q6: The Physically Interpretable Unified Action Space is innovative, but how does it handle robots with vastly different kinematics or action constraints?
>
> This is an insightful question. To handle robots with vastly different kinematics or action constraints, we need to feed the structure parameters of the robot arm into the model, although these parameters are not part of the dataset and are challenging to encode. Nevertheless, our action space ensures that the model learns shared physics and common end-effector patterns even without explicit structural parameters. This is achieved by aligning various physical quantities and maximizing the preservation of shared physical information across different robots.

---

> ### Comment · Reviewer_kiFc · 2024-11-26
>
> Thanks for the authors' clarification. In general, I think it's a good paper.

---

> > ### Author Response · Authors · 2024-11-27
> >
> > Thank you for your kind words and for your positive evaluation of our work. We truly appreciate your support and encouragement.
> >
> > We are also pleased to inform you that we have completed additional experiments addressing your question in Q2, and we have updated our response accordingly. The new results and analysis further reinforce the insights discussed in the paper.
> >
> > Thank you once again for your valuable feedback and thoughtful assessment, which has greatly contributed to improving the quality of our work.

---

> > ### Author Response · Authors · 2024-12-04
> > **Addtional Existing Simulation Benchmark of A Novel Single-Arm Franka Hardware**
> >
> > To further demonstrate the effectiveness of RDT, we conducted comparative experiments with other baseline methods using the ManiSkill simulation benchmark (an existing benchmark). Due to time and resource constraints, we selected the PickCube-v1 task for our experiments. The objective of PickCube-v1 is to grasp a red cube and move it to a specified target position. Detailed information about this task can be found in the [official ManiSkill documentation](https://maniskill.readthedocs.io/en/latest/tasks/table_top_gripper/index.html#pickcube-v1).
> > The simulator utilizes a single-arm Franka Panda robot, which differs from our current hardware setup. This variation allows us to assess the effectiveness of RDT across different hardware configurations, thereby demonstrating its versatility.
> >
> > ## Training Details
> >
> > ### Data
> >
> > Utilizing the [official ManiSkill repository](https://github.com/haosulab/ManiSkill), we generated 1,000 trajectories through motion planning. The initial action mode of these trajectories is absolute joint position control and we subsequently converted them into **delta end-effector pose** control to align with the pre-training action space of OpenVLA and Octo. We strictly adhered to the official codebases of OpenVLA and Octo, modifying only the dataset-loading scripts. Consequently, we finetuned OpenVLA and Octo using the delta end-effector pose data. For RDT and Diffusion-Policy we leverage joint position control data for training which is aligned with our pre-training stage as well.
> >
> > ### Training
> >
> > - OpenVLA is fine-tuned from the officially released pre-trained checkpoint with LoRA-rank 32 until converge, achieving a final training accuracy of 95%, which has reach the [success rate requirement](https://github.com/openvla/openvla) suggested by the authors. The loss curve and accuracy curve are at https://anonymous.4open.science/r/rdt-sim-eval-118C/openvla/openvla-training-acc.png
> > - Octo is fine-tuned from the officially released pre-trained checkpoint for 450k iterations until converge. The loss curve is at https://anonymous.4open.science/r/rdt-sim-eval-118C/octo/octo-training-loss.png
> > - Diffusion-Policy is trained from scratch for 1000 epochs. We select the checkpoint of 700 epoch which has the lowest validation sample loss of 1e-3.
> > - RDT is fine-tuned from our released pre-trained checkpoint for 150k iterations.
> >
> > ### Results
> >
> > Each method is evaluated over 250 trials (10 random seeds with 25 trials per seed). The quantitative results, including success rate mean and std value across 10 random seeds are presented below:
> >
> >
> > |              | RDT   | OpenVLA | Octo | Diffusion-Policy |
> > | ------------ | ----- | ------- | ---- | ---------------- |
> > | Success Mean | 72.4% | 20%     | 0%   | 44%              |
> > | Success Std  | ±2.8% | 0%      | 0%   | 0%               |

---

### Official Review · Reviewer_fucs · 2024-11-04

**Soundness:** 3
**Presentation:** 3
**Contribution:** 3
**Rating:** 6
**Confidence:** 4

**Summary:**

This paper presents an effort toward building a foundation model for bimanual manipulation. It proposes techniques to unify the action space, enabling training on a very large robot dataset. The authors also scale up the model to 1.2B parameters, making it the largest diffusion transformer model for robotics. In this process, the authors identify several key elements to improve training stability and performance. The resulting model achieves good zero-shot generalization performance on unseen and complex tasks.

**Strengths:**

This paper demonstrates strong performance in scaling up robotics models. It presents several interesting components that improve training stability and performance.

The unified action space, and especially the padding technique, is interesting.

The paper shows capabilities on several challenging real-world bimanual manipulation tasks.

**Weaknesses:**

Several claims are not very precise and not very clear. For example, the authors mention the nonlinearity and high frequency of robotic data. While it is true that the data is nonlinear, how does the proposed method tackle this challenge? The authors argue that changing the last linear layer to an MLP block solves this problem and brings significant performance improvements. While the performance is impressive, I think this requires more careful ablation experiments. Firstly, the entire diffusion transformer is already highly nonlinear due to stacking multiple layers, so why does adding more layers help in this case? Secondly, in the original UNet diffusion policy, the last layer is a Conv1dBlock; have the authors compared with that? Lastly, why is it only evaluated on the dexterity task?

It is also unclear how the model design choices address the “high frequency of robotic data.” Given that these two claims are highlighted in the abstract as the main challenges, I believe they require more careful analysis and discussion.

As the unified action representation is a major contribution of this paper, there should also be more analysis of this aspect. For example, what are the performance gains from using all the data because of the unified action space, compared to previous methods that use “robots with similar action spaces (Yang et al., 2023; Ghosh et al., 2023; Kim et al., 2024) or retain only a subset of inputs sharing the same structure (Collaboration et al., 2023; Yang et al., 2024)”? Additionally, what is the performance gain of the proposed padding strategy compared with padding with all zeros?

The authors argue the necessity of using RMSNorm and QKNorm, but they only show the loss without them (Figure 4(a)), which provides very little information on how effective the proposed approach is and whether it addresses the instability issue. It also does not mention how to integrate these normalization layers within the transformer.

Scaling up the models and data is certainly attractive, and the paper shows impressive results. However, most of the analysis are “binary”, which means the results are either with or without. Showing more datapoints (model with different size, using different percentage of the dataset) will present more insights to the reader.

**Questions:**

My questions are mainly also discussed in the weakness. Here is an summary:
1. Given that the diffusion transformer already contains multiple nonlinear layers, why does adding an additional layer improve performance? Have you conducted ablation studies to support this choice?
2. In the original UNet diffusion policy, the last layer is a Conv1dBlock. Have you compared the performance of your MLP block with this alternative?
3. What is the the MLP block performance gains on other tasks?
4. Can you provide more insight into how the model design specifically addresses high frequency change?
5. What are the performance gains from using all data through this unified action space compared to previous methods that only use robots with similar action spaces or retain subsets of inputs with a common structure?
6. How does the proposed padding strategy improve performance compared to zero padding? Could you provide comparative results?
7. How do these normalization layers integrate into the transformer architecture, and how do they address instability? Have you tested different configurations of these layers?

---

> ### Author Response · Authors · 2024-11-18
> **Thank you for the valuable review (1/3)**
>
> We gratefully appreciate the time and effort you have dedicated to reviewing our manuscript. Your insightful feedback has greatly improved our work and inspired a deeper exploration of our model architecture.
>
> In this revision, we have thoroughly addressed each of your comments and questions. Below, we provide detailed responses to your points to clarify your concerns and further strengthen our work.
>
> ## Q1: Given that the diffusion transformer already contains multiple nonlinear layers, why does adding an additional layer improve performance? Have you conducted ablation studies to support this choice?
>
> First, we would like to humbly emphasize that the MLPs in Transformer layers are different from the MLP as a decoder (in the final layer). The former is a function on the latent space, while the latter is a mapping from the high-dimensional latent space back to the low-dimensional physical space. Thus, politely, we do not suggest considering the decoder as "an additional layer".
>
> Next, let's explain why nonlinear decoders are beneficial. If the decoder is linear, then we can easily prove that the nonlinearity of the physical space is consistent with the latent space. If we use a nonlinear decoder (such as MLP), then it is possible to greatly weaken the nonlinearity when coming to the latent space, thereby allowing the model to better fit nonlinear actions in the latent space. The idea of using nonlinear mappings between low-dimensional nonlinear space and high-dimensional space is very common. This often effectively decreases nonlinearity, and even only linear layers are needed to fit in high-dimensional latent space (the Koopman operators are classic examples [*1-*3]).
>
> Finally, we respectfully note that we did conduct ablation studies to support our choice. Please refer to Fig. 4(b) in our manuscript, where the experiments show that when the decoder is a linear layer, the model will have a significant performance drop on dexterity tasks.
>
> [*1] Mauroy, A., Susuki, Y., \& Mezic, I. (2020). Koopman operator in systems and control. Berlin, Germany: Springer International Publishing.
>
> [*2] Korda, M., \& Mezić, I. (2018). Linear predictors for nonlinear dynamical systems: Koopman operator meets model predictive control. Automatica, 93, 149-160.
>
> [*3] Lusch, B., Kutz, J. N., \& Brunton, S. L. (2018). Deep learning for universal linear embeddings of nonlinear dynamics. Nature communications, 9(1), 4950.
>
> ## Q2: In the original UNet diffusion policy, the last layer is a Conv1dBlock. Have you compared the performance of your MLP block with this alternative?
>
> We modestly think that Conv1dBlock is usually used with CNN-based architectures such as UNet, but not with Transformer. This is because the self-attention layer in the Transformer already includes convolution in terms of expressiveness. You can see that in the original Transformer diffusion policy, the last layer, or equivalently, the decoder is a linear layer instead of Conv1dBlock. Since our backbone is the Transformer, we mainly compare our MLP decoder with the mainstream choice -- the linear decoder.

---

> ### Author Response · Authors · 2024-11-18
> **Thank you for the valuable review (2/3)**
>
> ## Q3: What are the MLP block performance gains on other tasks?
>
> To comprehensively evaluate the advantage of employing an MLP decoder, we conduct three additional ablation experiments in this response. To ensure the fairness and reduce the resources needed by pre-training, all the RDT variants are trained directly on the fine-tuning dataset without large-scale pre-training.
>
> Here are results of the original experiment in Fig. 4(b).
>
> Robot Dog (**Dexterity**):
>
> | Method      | Grab Remote | Push Joystick | Total | Walk Straight |
> | ----------- | ----------- | ------------- | ----- | ------------- |
> | RDT w/o MLP | 92          | 48            | 48    | 12            |
> | RDT         | 100         | 64            | 64    | 32            |
> | Gain        | +8          | +16           | +16   | +20           |
>
> Here are results of three additional experiments.
>
> Pour Water-L-1/3 (**Instruction Following**):
>
> | Method      | Pick Up Bottle | Pour Water | Place Back Bottle | Total | Correct Hand | Correct Amount |
> | ----------- | -------------- | ---------- | ----------------- | ----- | ------------ | -------------- |
> | RDT w/o MLP | 100            | 50         | 50                | 50    | 100          | 37.5           |
> | RDT         | 100            | 75         | 62.5              | 62.5  | 100          | 62.5           |
> | Gain        | 0              | +25        | +12.5             | +12.5 | 0            | +25            |
>
> Pour Water: unseen room 2 (**Unseen Scene**):
>
> | Method      | Pick Up Bottle | Pour Water | Place Back Bottle | Total |
> | ----------- | -------------- | ---------- | ----------------- | ----- |
> | RDT w/o MLP | 100            | 50         | 50                | 50    |
> | RDT         | 100            | 87.5       | 75                | 75    |
> | Gain        | 0              | +37.5      | +25               | +25   |
>
> Fold Shorts (**1-Shot Learning**):
>
> | Method      | Total |
> | ----------- | ----- |
> | RDT w/o MLP | 32    |
> | RDT         | 40    |
> | Gain        | +8    |
>
> Through the above results, we can conclude that replacing the linear decoder with an MLP decoder is beneficial, which supports our analysis in **Q1**.

---

> ### Author Response · Authors · 2024-11-18
> **Thank you for the valuable review (3/3)**
>
> ## Q4: Can you provide more insight into how the model design specifically addresses high-frequency change?
>
> We agree that more explanation is helpful for readers to understand our design choice. According to the neural tangent kernel (NTK) theory [*4], we know that neural networks learn different frequencies at different speeds: they learn low-frequency components faster and fail to capture high-frequency behaviors. This causes the neural network to learn smooth actions and fail to represent fine-grained actions (such as aiming, twisting, and plugging). These fine-grained operations are crucial for dexterous tasks. To solve this problem, we introduce an MLP encoder with Fourier Features of different frequencies [*4] so that the model can learn high-frequency components more effectively.
>
> [*4] Tancik, M., Srinivasan, P., Mildenhall, B., Fridovich-Keil, S., Raghavan, N., Singhal, U., ... \& Ng, R. (2020). Fourier features let networks learn high-frequency functions in low-dimensional domains. Advances in neural information processing systems, 33, 7537-7547.
>
> ## Q5: What are the performance gains from using all data through this unified action space compared to previous methods that only use robots with similar action spaces or retain subsets of inputs with a common structure?
>
> We carefully point out that, among our baselines, OpenVLA is a representative of this type of method. Its action space only has EEF: for datasets that contain both EEF and joints, it removes the joints part; for datasets that only have joints, it removes the entire dataset. As can be seen from Fig. 1 and Table 3, our method has an average gain of more than 60% over it. This is because we can leverage more data with the help of the unified action space, achieving better generalizability.
>
> ## Q6: How does the proposed padding strategy improve performance compared to zero padding?
>
> We'd like to sincerely clarify that the zero padding technique is a part of our strategy. For a specific robot, we fill the physical quantities in its original action space into the corresponding positions in the unified action space (see the left side of Fig. 3), and pad the remaining unavailable positions with 0. In addition, we also feed in a 0-1 indicator vector to indicate whether each position in the unified action space is valid or not. This is to distinguish between the padded 0 and the physical 0 value that appears in the original action. For more details, please refer to the second paragraph of Appendix C and refer to Appendix F.
>
>
> ## Q7: Showing more data points (models with different sizes, using different percentages of the dataset) will present more insights to the reader.
>
> We appreciate the reviewer’s feedback and understand the desire for a more granular analysis across different model sizes and dataset percentages. However, due to the substantial computational cost and time constraints, conducting such experiments for additional data points was not feasible within the scope of this work. Pre-training large models, such as the 1B parameter model, requires significant GPU resources and time, while real-world robot experiments demand extensive testing (often requiring dozens of trials per task). Given these limitations, we believe that the binary experiments presented—comparing pre-trained vs. non-pre-trained models and large vs. small models—sufficiently highlight our key contributions. Furthermore, we acknowledge the importance of scaling laws and plan to explore this in future work, once additional funding/resources become available. We hope the reviewer can understand that while more data points would indeed provide valuable insights, the current analysis remains a strong demonstration of the major findings.
>
> ## Q8: How do these normalization layers integrate into the transformer architecture, and how do they address instability? Have you tested different configurations of these layers?
>
> Thank you for your insightful advice. For RMSNorm, we replace each LayerNorm with RMSNorm in the original Transformer architecture. For QKNorm, we normalize query and key vectors before calculating attention in each attention layer. We refer to Appendix B for more details. In our revised manuscript, we compare the loss curves with and without our improved normalization technique in Figure 4(a). We can see that our normalization technique has improved the training stability and avoided the loss explosion. Besides, as discussed in **Q7**, we believe our current ablation study in Figure 4(a) can highlight the core contribution of our improved normalization technique given limited resources. In future work, once additional resources are ready, we would like to test different configurations of these layers to get more insights.

---

> ### Author Response · Authors · 2024-11-23
> **Look forward to further feedback**
>
> Dear Reviewer fucs,
>
> We are grateful for the thoughtful and constructive feedback you have provided. We have carefully considered each of your comments and have made the necessary revisions to our manuscript. We hope you will find the response satisfactory and are eager to receive your further feedback.
>
> Best, Authors

---

> > ### Comment · Reviewer_fucs · 2024-11-25
> >
> > Thank you for your detailed and valuable feedback. Many of my concerns are addressed. I still believe it will be better to provide more evaluation datapoint but I understand it is challenging given the limited time and resources requirements. I will keep my original score as weak accept.

---

> > > ### Author Response · Authors · 2024-11-27
> > >
> > > Thank you for your thoughtful follow-up and for acknowledging the challenges associated with providing more evaluation data points. We deeply appreciate your feedback and your overall positive opinion of our work.
> > >
> > > We would like to reiterate that the primary focus of our paper is on developing a generalized, state-of-the-art foundational model for bimanual manipulation, a domain that is significantly more challenging than the single-arm tasks emphasized in previous works such as OpenVLA and Octo. Our experiments demonstrate the exceptional generalization ability of our model on bimanual manipulation tasks, a critical novelty that sets our work apart from prior approaches.
> > >
> > > While scaling laws and additional data points are indeed valuable for understanding broader trends, they are not the central theme of our study. Addressing these points in detail would require substantial time and resources, far exceeding the scope of this work. Furthermore, we believe the binary comparisons provided (e.g., pre-trained vs. non-pre-trained models, and large vs. small models) sufficiently highlight our key contributions and effectively validate the strength of our approach within the context of bimanual manipulation.
> > >
> > > We hope this clarification aligns with your understanding, and we sincerely thank you for your valuable time and for recognizing the inherent challenges.

---

> > > ### Author Response · Authors · 2024-12-04
> > > **Addtional Existing Simulation Benchmark of A Novel Single-Arm Franka Hardware**
> > >
> > > To further demonstrate the effectiveness of RDT, we conducted comparative experiments with other baseline methods using the ManiSkill simulation benchmark (an existing benchmark). Due to time and resource constraints, we selected the PickCube-v1 task for our experiments. The objective of PickCube-v1 is to grasp a red cube and move it to a specified target position. Detailed information about this task can be found in the [official ManiSkill documentation](https://maniskill.readthedocs.io/en/latest/tasks/table_top_gripper/index.html#pickcube-v1).
> > > The simulator utilizes a single-arm Franka Panda robot, which differs from our current hardware setup. This variation allows us to assess the effectiveness of RDT across different hardware configurations, thereby demonstrating its versatility.
> > >
> > > ## Training Details
> > >
> > > ### Data
> > >
> > > Utilizing the [official ManiSkill repository](https://github.com/haosulab/ManiSkill), we generated 1,000 trajectories through motion planning. The initial action mode of these trajectories is absolute joint position control and we subsequently converted them into **delta end-effector pose** control to align with the pre-training action space of OpenVLA and Octo. We strictly adhered to the official codebases of OpenVLA and Octo, modifying only the dataset-loading scripts. Consequently, we finetuned OpenVLA and Octo using the delta end-effector pose data. For RDT and Diffusion-Policy we leverage joint position control data for training which is aligned with our pre-training stage as well.
> > >
> > > ### Training
> > >
> > > - OpenVLA is fine-tuned from the officially released pre-trained checkpoint with LoRA-rank 32 until converge, achieving a final training accuracy of 95%, which has reach the [success rate requirement](https://github.com/openvla/openvla) suggested by the authors. The loss curve and accuracy curve are at https://anonymous.4open.science/r/rdt-sim-eval-118C/openvla/openvla-training-acc.png
> > > - Octo is fine-tuned from the officially released pre-trained checkpoint for 450k iterations until converge. The loss curve is at https://anonymous.4open.science/r/rdt-sim-eval-118C/octo/octo-training-loss.png
> > > - Diffusion-Policy is trained from scratch for 1000 epochs. We select the checkpoint of 700 epoch which has the lowest validation sample loss of 1e-3.
> > > - RDT is fine-tuned from our released pre-trained checkpoint for 150k iterations.
> > >
> > > ### Results
> > >
> > > Each method is evaluated over 250 trials (10 random seeds with 25 trials per seed). The quantitative results, including success rate mean and std value across 10 random seeds are presented below:
> > >
> > >
> > > |              | RDT   | OpenVLA | Octo | Diffusion-Policy |
> > > | ------------ | ----- | ------- | ---- | ---------------- |
> > > | Success Mean | 72.4% | 20%     | 0%   | 44%              |
> > > | Success Std  | ±2.8% | 0%      | 0%   | 0%               |

---

### Official Review · Reviewer_VxiE · 2024-11-06

**Soundness:** 3
**Presentation:** 3
**Contribution:** 4
**Rating:** 8
**Confidence:** 4

**Summary:**

This paper develops a 1.2B-parameter robotics foundation model that is trained and evaluated on real robot data for bimanual manipulation. The model is trained with imitation learning: (i) pretrained on 1M trajectories combining available datatsets collected for different robots, and (ii) fine-tuned on a self-collected dataset with 6k demonstrations for a Mobile Aloha robot. The model adopts a diffusion transformer (DiT) architecture that takes multi-modal inputs (images, language, etc.) and generate action chunks with multimodal distribution.

The model is evaluated on 7 real robot tasks against mainstream baselines. The comparison shows that the model can: (i) generalize zero-shot to novel objects, scenes and language, (ii) learn new skills with few data, (iii) accomplish dexterous tasks. Ablation studies show that larger model and pretraining with large data significantly boost the performance.

**Strengths:**

The paper presents a complete and remarkable research work that pushes forward the boundary of large-scale robot learning.
- The model is developed on top of the diffusion transformer with a unified action space, which allows large-scale pretraining on heterogeneous robot data to boost the performance
- The authors collect the largest robot dataset for bimanual manipulation with comprehensive task coverage for fine-tuning the model
- The experiments show that the advantage of the model from a foundation model aspect: generalization, few-shot learning, and scaling behavior

**Weaknesses:**

- While the paper demonstrates that the foundation model is allows zero-shot and few-shot generalization, and can achieve dexterous manipulation, each of these characteristics is only validated on ~one task and may be insufficient. Evaluations on more tasks and existing benchmark tasks will complete the results.
- It seems that the baselines are not trained on the complete fine-tuning dataset. This doesn't form an apple-to-apple comparison.
- The writing of the paper has room for improvement. Some of the sentences are too long, which prevent reading of the paper smoothly.
- Mistakes in citing papers. For example, it seems that "Xie et al. 2020" is wrongly used as the reference of DiT in line 83 and 319.

**Questions:**

- The model uses frozen vision encoders. Does it mean existing pretrained visual representation is sufficient for robot manipulation? I wonder if the authors spot any cases where pretrained visual encoder is insufficient and lead to unsatisfactory performance.
- In pretraining on heterogeneous data with varied control frequency, the model takes control frequency as conditioning. I wonder how this strategy works in practice - does the model learns policy prior corresponding to different control frequency?

---

> ### Author Response · Authors · 2024-11-18
> **Thank you for the valuable review (1/3)**
>
> We sincerely thank you for your time and effort in reviewing our manuscript. Your insightful feedback has been invaluable in improving our work, and we appreciate your suggestion for additional experiments to gain more insights.
>
> In this revised submission, we have carefully addressed all your comments and questions, including the requested experiments. Below, we provide detailed responses to each of your points to clarify and further strengthen our work.
>
> ## Q1: Evaluations on more tasks and existing benchmark tasks will complete the results.
>
> Thank you for your valuable feedback and insightful advice. We have expanded our evaluation to cover a broader range of real-world tasks, which are even more challenging. Given the constraints of time and resources, we designed four additional experiments from three dimensions: **Unseen Scene**, **Unseen Object**, and **Instruction Following**.
>
> ### Task Definition
>
> Below, we provide an overview of each task.
>
> | **TASK NAME**                   | **DIMENSION**         | **EXPLANATION**                                              |
> | ------------------------------- | --------------------- | ------------------------------------------------------------ |
> | Stack Tomato Can (STC)          | Unseen Scene          | To stack the right tomato can onto the left one (in seen and unseen scenes) |
> | Stack Tomato Can Mahjong (STCM) | Unseen Object         | To stack the tomato can onto the (unseen) foam mahjong       |
> | Put Orange Lunchbox (POL)       | Instruction Following | To pick up the orange and place it into the paper lunchbox (where the robot needs to distinguish between orange and apple) |
> | Put Apple Lunchbox (PAL)        | Instruction Following | To pick up the green apple and place it into the paper lunchbox |
>
> Task visualizations can be found at https://anonymous.4open.science/r/rdt-demo-543D/addtional_experiment_visualization.png
>
> ### Implementation Details
>
> The implementation details for each method are described as follows:
>
> **1. Training Details for ACT**
>
> ACT was fully trained using our finetune dataset. Since ACT does not support language instructions as input, it was excluded from experiments involving **Instruction Following** tasks.
>
> **2. Training and Fine-Tuning Details for OpenVLA**
>
> - **Pretraining**: We continued pretraining OpenVLA on an integrated bimanual dataset (a combination of Mobile ALOHA dataset and our finetune dataset).
> - **Fine-tuning**: We fine-tuned OpenVLA on the entire fine-tuning dataset. The model achieved an average classification accuracy of **30%**.
>
>
> **3. Training and Fine-Tuning Details for Octo**
>
> - **Pretraining**: Octo was pre-trained on the same bimanual dataset as OpenVLA.
> - **Fine-tuning**: We fine-tuned Octo on the entire fine-tuning dataset. The model achieved a validation set loss of approximately **5e-2**.
>
> **4. Implementation of RDT**
>
> For RDT, we utilized our previously fine-tuned checkpoints in all experiments, consistent with prior evaluations.
>
> ### Experiment Results
>
> Each task was evaluated over **8 trials**. The quantitative results are shown below:
>
> |               | **STC (Seen)** |      | **STC (Unseen 1)** |      | **STC (Unseen 2)** |      | **STCM** |      | **POL** |      | **PAL** |      |
> | ------------- | -------------- | ---- | ------------------ | ---- | ------------------ | ---- | -------- | ---- | ------- | ---- | ------- | ---- |
> | **ACT**       | 62.5           | 37.5 | 0                  | 0    | 0                  | 0    | 50       | 0    | -       | -    | -       | -    |
> | **OpenVLA**   | 12.5           | 0    | 0                  | 0    | 0                  | 0    | 0        | 0    | 0       | 0    | 0       | 0    |
> | **Octo**      | 0              | 0    | 0                  | 0    | 0                  | 0    | 0        | 0    | 0       | 0    | 0       | 0    |
> | **RDT(Ours)** | 75             | 50   | 75                 | 37.5 | 62.5               | 25   | 75       | 50   | 87.5    | 87.5 | 75      | 75   |
> ### Results Analysis
>
> We highlight the challenges of each task and evaluate method performance. **Stack Tomato Can (STC)** involves adapting to new environments and precise placement of the cylindrical, slippery tomato can. **Stack Tomato Can Mahjong (STCM)** requires generalization to unseen foam mahjong pieces.  **Put Orange Lunchbox (POL)** and **Put Apple Lunchbox (PAL)** demand instruction-following to distinguish and pick the correct object.
>
> In the seen scene, ACT can grasp and occasionally place the tomato can but fails in the unseen scene, showing erratic movements. Octo reliably reaches targets but fails to grasp due to ineffective gripper closure. OpenVLA can grasp in the seen scene but struggles with placement and frequently misidentifies objects, while also failing to generalize to unseen scenes.
>
> In contrast, RDT demonstrates robust generalization, successfully handling unseen objects and scenes while following instructions accurately.

---

> ### Author Response · Authors · 2024-11-18
> **Thank you for the valuable review (2/3)**
>
> ## Q2: It seems that the baselines are not trained on the complete fine-tuning dataset.
>
> We want to humbly point out that we have carefully fine-tuned OpenVLA and Octo on the complete multi-task fine-tuning dataset, adhering closely to the settings described in their respective original papers. Despite these efforts, we observed that both models failed to converge to a deployable loss level, resulting in random or meaningless patterns of movement in real-world scenarios. Consequently, we opted to fine-tune the models on specific tasks individually. Further details on our implementation can be found in App. H.
>
> Reviewer DG58 proposed a valuable hypothesis that these models, being primarily pre-trained on dual-arm robot datasets, may face a significant domain gap that hinders successful adaptation to bi-manual manipulation. To investigate this, we extended the pre-training of the officially released models using open-source bi-manual datasets, including ALOHA (static and mobile) and our fine-tuning dataset, totaling over 7K episodes. However, even after this additional pre-training, the models still failed to converge during fine-tuning across the entire dataset, as depicted in Table a.
>
> *Table a: The performance comparison of different methods further pre-trained on bimanual dataset (ALOHA Static and ALOHA Mobile and our finetune dataset). They are further finetuned with whole fine-tuning dataset.*
>
> *Wash Cup: seen cup 1 | unseen cup 1 | unseen cup 2 (Unseen Object)*
>
> |          | Pick Up Cup |      |      | Turn On Faucet |      |      | Get Water |      |      | Pour Out Water |      |      | Place Back Cup |      |      | Total |      |      |
> | -------- | ----------- | ---- | ---- | -------------- | ---- | ---- | --------- | ---- | ---- | -------------- | ---- | ---- | -------------- | ---- | ---- | ----- | ---- | ---- |
> | Octo     | 0           | 0    | 0    | 0              | 0    | 0    | 0         | 0    | 0    | 0              | 0    | 0    | 0              | 0    | 0    | 0     | 0    | 0    |
> | OpenVLA* | 0           | 0    | 0    | 0              | 0    | 0    | 0         | 0    | 0    | 0              | 0    | 0    | 0              | 0    | 0    | 0     | 0    | 0    |
>
>  *Pour Water-L-1/3 | Pour Water-R-2/3 (**Instruction Following**)*
>
> |         | Pick Up Bottle |      |      | Pour Water |      |      | Place Back Bottle |      |      | Total |      |      | Correct Hand |      |      | Correct Amount |      |
> | ------- | -------------- | ---- | ---- | ---------- | ---- | ---- | ----------------- | ---- | ---- | ----- | ---- | ---- | ------------ | ---- | ---- | -------------- | ---- |
> | Octo    | 0             | 0    | 0    | 0          | 0    | 0    | 0                 | 0    | 0    | 0     | 0    | 0    | 0           | 0    | 0    | 0              | 0    |
> | OpenVLA | 50              | 0    | 0    | 0          | 0    | 0    | 0                 | 0    | 0    | 0     | 0    | 0    | 50           | 0    | 0    | 0              | 0    |
>
>  *Pour Water: unseen room 1 | unseen room 2 | unseen room 3 (**Unseen Scene**)*
>
> |         | Pick Up Bottle |      |      | Pour Water |      |      | Place Back Bottle |      |      | Total |      |      |
> | ------- | -------------- | ---- | ---- | ---------- | ---- | ---- | ----------------- | ---- | ---- | ----- | ---- | ---- |
> | Octo    | 0              | 0    | 0    | 0          | 0    | 0    | 0                 | 0    | 0    | 0     | 0    | 0    |
> | OpenVLA | 0              | 0    | 0    | 0          | 0    | 0    | 0                 | 0    | 0    | 0     | 0    | 0    |
>
>  *Fold Shorts (**1-Shot**)*
>
> |         | Total |
> | ------- | ----- |
> | Octo    | 0     |
> | OpenVLA | 0     |
>
>  *Handover (**5-Shot**)*
>
>
> |         | Pick Up Pen | Switch Hand | Drop Pen | Fall into Box | Total |
> | ------- | ----------- | ----------- | -------- | ------------- | ----- |
> | Octo    | 0           | 0           | 0        | 0             | 0     |
> | OpenVLA | 0           | 0           | 0        | 0             | 0     |
>
> In summary, despite extensive efforts in re-implementing and fine-tuning these models for optimal performance, the issue of insufficient convergence persists. For OpenVLA, there might be a broader scientific challenge in scaling vision-language-action (VLA) models effectively for **bi-manual manipulation**, which could be a possible reason why they do not consider bi-manual manipulation [*1]. For Octo, other researchers have also reported the phenomenon of non-convergence and random behaviors of the robot when adapting Octo to bi-manual manipulation [*2]. We can conclude that bimanual manipulation is challenging for previous foundation models, which is addressed by designing a unified action space and a powerful architecture in our RDT. While beyond the scope of this work, we believe adapting VLA to bimanual manipulation will be a promising direction.
>
> [*1] https://github.com/openvla/openvla/issues/48
>
> [*2] https://github.com/octo-models/octo/issues/29

---

> ### Author Response · Authors · 2024-11-18
> **Thank you for the valuable review (3/3)**
>
> ## Q3: Some of the sentences are too long.
>
> Thanks for pointing it out. We have made some long sentences shorter, thus improving the readability. See our revised manuscript for modifications (highlighted in blue).
>
> ## Q4: Mistakes in citing papers.
>
> We thank the reviewer so much for pointing this out. We have corrected the wrong citations, as shown in lines 83 and 319 of the revised manuscript (highlighted in blue).
>
> ## Q5: Why does the model use frozen vision encoders?
>
> You raise a very good question, which we may have forgotten to explain in the manuscript. We think that it is better to freeze the weights of the vision encoder. The frozen vision encoder can retain the robust, generalizable features acquired through extensive Internet-scale pre-training. In contrast, fine-tuning the encoder on much smaller and domain-specific robotic datasets may cause overfitting, thereby learning less generalizable features. Similar observations have also been documented in video generation models (See e.g., [*1]).
>
> [*1] Polyak, A., Zohar, A., Brown, A., Tjandra, A., Sinha, A., Lee, A., ... \& Du, Y. (2024). Movie gen: A cast of media foundation models. arXiv preprint arXiv:2410.13720.
>
> ## Q6: How does the strategy of taking control frequency as conditioning work in practice?
>
> Unlike video data, the frame rate (or equivalently, control frequency) of robotic data is widely distributed. In our case, the control frequency is almost evenly distributed from 1Hz to 66Hz. It is unreasonable to simply align all datasets to 1Hz and it is unacceptable for some robots that need to react quickly. Therefore, we adopt the conditioning in RDT and feed the control frequency into the model as an important context, as illustrated in Fig. 3. In this way, RDT knows whether the action sequence to generate is dense in time (i.e., high control frequency) or sparse in time (i.e., low control frequency). Without such context, the entire action prediction problem will be ill-defined.

---

> ### Comment · Reviewer_VxiE · 2024-11-25
>
> I would thank the authors for the clarifications and additional experiments despite a tight timeframe. I think it's overall a good paper and I'm happy to accept it. I have updated my score.

---

> > ### Author Response · Authors · 2024-11-27
> > **Thank you for your thoughtful reply and for considering our clarifications.**
> >
> > We are pleased to inform you that we have completed the additional experiments and updated our responses to Q1 and Q2 accordingly. We kindly invite you to review the revised sections, and we look forward to your feedback.

---

> > ### Author Response · Authors · 2024-12-04
> > **Addtional Existing Simulation Benchmark of A Novel Single-Arm Franka Hardware**
> >
> > To further demonstrate the effectiveness of RDT, we conducted comparative experiments with other baseline methods using the ManiSkill simulation benchmark (an existing benchmark). Due to time and resource constraints, we selected the PickCube-v1 task for our experiments. The objective of PickCube-v1 is to grasp a red cube and move it to a specified target position. Detailed information about this task can be found in the [official ManiSkill documentation](https://maniskill.readthedocs.io/en/latest/tasks/table_top_gripper/index.html#pickcube-v1).
> > The simulator utilizes a single-arm Franka Panda robot, which differs from our current hardware setup. This variation allows us to assess the effectiveness of RDT across different hardware configurations, thereby demonstrating its versatility.
> >
> > ## Training Details
> >
> > ### Data
> >
> > Utilizing the [official ManiSkill repository](https://github.com/haosulab/ManiSkill), we generated 1,000 trajectories through motion planning. The initial action mode of these trajectories is absolute joint position control and we subsequently converted them into **delta end-effector pose** control to align with the pre-training action space of OpenVLA and Octo. We strictly adhered to the official codebases of OpenVLA and Octo, modifying only the dataset-loading scripts. Consequently, we finetuned OpenVLA and Octo using the delta end-effector pose data. For RDT and Diffusion-Policy we leverage joint position control data for training which is aligned with our pre-training stage as well.
> >
> > ### Training
> >
> > - OpenVLA is fine-tuned from the officially released pre-trained checkpoint with LoRA-rank 32 until converge, achieving a final training accuracy of 95%, which has reach the [success rate requirement](https://github.com/openvla/openvla) suggested by the authors. The loss curve and accuracy curve are at https://anonymous.4open.science/r/rdt-sim-eval-118C/openvla/openvla-training-acc.png
> > - Octo is fine-tuned from the officially released pre-trained checkpoint for 450k iterations until converge. The loss curve is at https://anonymous.4open.science/r/rdt-sim-eval-118C/octo/octo-training-loss.png
> > - Diffusion-Policy is trained from scratch for 1000 epochs. We select the checkpoint of 700 epoch which has the lowest validation sample loss of 1e-3.
> > - RDT is fine-tuned from our released pre-trained checkpoint for 150k iterations.
> >
> > ### Results
> >
> > Each method is evaluated over 250 trials (10 random seeds with 25 trials per seed). The quantitative results, including success rate mean and std value across 10 random seeds are presented below:
> >
> >
> > |              | RDT   | OpenVLA | Octo | Diffusion-Policy |
> > | ------------ | ----- | ------- | ---- | ---------------- |
> > | Success Mean | 72.4% | 20%     | 0%   | 44%              |
> > | Success Std  | ±2.8% | 0%      | 0%   | 0%               |

---

> > ### Author Response · Authors · 2024-12-04
> > **Thanks for the update**
> >
> > Thank you very much for increasing the rating. We'll try our best to further improve the paper in the final version.

---

### Author Response · Authors · 2024-11-18
**Manuscript Update Note**

We thank each reviewer for their invaluable feedback. We have carefully considered their suggestions, addressed their concerns, and revised our manuscript accordingly. All changes are highlighted in **blue** in the revised manuscript. The details of these modifications are as follows:

1. Shorten some long sentences and improve the readability.
2. Fix some wrong citations.
3. Add a loss curve with our improved normalization technique in Figure 4(a).
4. Fix some typos.

---

### Meta-Review · Area_Chair_h3XV · 2024-12-16

**Metareview:**

The paper introduces RDT (Robotics Diffusion Transformer), a diffusion-based foundation model for bimanual manipulation. The key scientific claims and findings include:
+ A novel vision-language-action (VLA) architecture based on diffusion transformers that effectively handles multi-modal inputs and complex action distributions for bimanual tasks
+ A Physically Interpretable Unified Action Space that enables training on heterogeneous robot datasets while preserving physical meaning
+  Successful scaling to 1.2B parameters through pre-training on the largest collection of multi-robot datasets to date
+ Strong empirical results demonstrating zero-shot generalization, few-shot learning, and instruction following capabilities on real robot hardware

Weaknesses:
- More analysis could be provided on limitations of the unified action representation and potential failure cases
- Additional details on scaling behavior and model variants would strengthen empirical validation
- Comparison with baselines could be more equitable in terms of training data and protocols
- Some implementation details of the architecture could be better clarified

The clear strengths in terms of novel technical approach, strong empirical results, and potential impact on the field outweigh the limitations. I recommend accepting this paper.

**Additional Comments On Reviewer Discussion:**

Initial concerns focused on:
- Fairness of baseline comparisons, particularly regarding training data (Reviewer dG58)
- Technical details of the architecture and representation (Reviewer fucs)
- Evaluation scope and real-world validation (Reviewer VxiE)
- Clarity of experimental protocols (Reviewer kiFc)

The authors provided thorough responses and additional experiments:
- Extended evaluation on diverse real-world tasks and an existing benchmark
- Clarified architectural details and motivation for key design choices
- Performed ablation studies isolating the contribution of different components
- Added comprehensive analysis of baseline comparison protocols
- Improved clarity of experimental methodology

The responses effectively addressed the key concerns. All reviewers ultimately recommended acceptance given the strong results and broader impact potential of the work.

---

### Decision · Program_Chairs · 2025-01-22

Accept (Poster)